# Quantum key distribution implemented with *d*-level time-bin entangled photons

Hao Yu[1,2,12], Stefania Sciara [1,12] ✉, Mario Chemnitz [1,3,4], Nicola Montaut[1], Benjamin Crockett[1], Bennet Fischer [1,3], Robin Helsten[1], Benjamin Wetzel [5], Thorsten A. Goebel[4,6], Ria G. Krämer[4], Brent E. Little[7], Sai T. Chu [8], Stefan Nolte [4,6], Zhiming Wang [2], José Azaña [1], William J. Munro [9], David J. Moss [10,11] & Roberto Morandotti [1] ✉

High-dimensional photon states (qudits) are pivotal to enhance the information capacity, noise robustness, and data rates of quantum communications. Time-bin entangled qudits are promising candidates for implementing high-dimensional quantum communications over optical fiber networks with processing rates approaching those of classical telecommunications. However, their use is hindered by phase instability, timing inaccuracy, and low scalability of interferometric schemes needed for time-bin processing. As well, increasing the number of time bins per photon state typically requires decreasing the repetition rate of the system, affecting in turn the effective qudit rates. Here, we demonstrate a fiber-pigtailed, integrated photonic platform enabling the generation and processing of picosecond-spaced time-bin entangled qudits in the telecommunication C band via an on-chip interferometry system. We experimentally demonstrate the Bennett-Brassard-Mermin 1992 quantum key distribution protocol with time-bin entangled qudits and extend it over a 60 km-long optical fiber link, by showing dimensionality scaling without sacrificing the repetition rate. Our approach enables the manipulation of time-bin entangled qudits at processing speeds typical of standard telecommunications (10 s of GHz) with high quantum information capacity per single frequency channel, representing an important step towards an efficient implementation of high-data rate quantum communications in standard, multi-user optical fiber networks.

One of the most compelling and immediate applications of quantum photonics is associated with quantum communications, which have the ability to deliver enhanced security compared to their classical counterparts[1–3]. Protocols such as quantum key distribution (QKD) facilitate the exchange of (theoretically secure) secret keys between two distant clients—Alice and Bob. Among the many implementations of QKD, entanglement-based schemes provide the means to enable device-independent protocols and source-independent security,

[1]Institut national de la recherche scientifique—Centre Énergie Matériaux Télécommunications, Varennes, QC, Canada. [2]Shimmer Center, Tianfu Jiangxi Laboratory, Chengdu, China. [3]Leibniz Institute of Photonic Technology, Jena, Germany. [4]Friedrich Schiller University Jena, Abbe Center of Photonics, Institute of Applied Physics, Jena, Germany. [5]Xlim Research Institute, CNRS UMR 7252, University of Limoges, Limoges, France. [6]Fraunhofer Institute for Applied Optics and Precision Engineering IOF, Center of Excellence in Photonics, Jena, Germany. [7]QXP Technology Inc., Xi'an, China. [8]Department of Physics, City University of Hong Kong, Kowloon Tong, Hong Kong. [9]Okinawa Institute of Science and Technology Graduate University, Okinawa, Japan. [10]Optical Sciences Centre, Swinburne University of Technology, Hawthorn, VIC, Australia. [11]ARC Centre of Excellence in Optical Microcombs for Breakthrough Science (COMBS), Melbourne, VIC, Australia. [12]These authors contributed equally: Hao Yu, Stefania Sciara. ✉e-mail: stefania.sciara@inrs.ca; roberto.morandotti@inrs.ca

meaning that neither Alice or Bob needs to hold the photon source to have a secure communication[2,4,5]. This property relies on the quantum mechanical completeness of entanglement; Alice and Bob can verify that their photons are entangled based solely on their own measurements, without needing any information from a third party controlling the source.

To foster the implementation of a highly secure quantum internet, much interest has grown in realizing entanglement distribution systems over optical fiber networks that are compatible with standard telecommunications infrastructure[6–8]. However, entanglement-distribution schemes based on two-level photonic systems (qubits) are fragile to noise and losses and suffer from low effective key generation rates[9,10].

High-dimensional photon states—$d$-level or qudits—hold the key to overcoming these limitations, as they can increase the information capacity, noise robustness, distance, and secret key rates of QKD implementations[11–14]. Energy-time entanglement[15,16] and its discrete form, also known as time-bin entanglement[17–19], are promising candidates towards realizing high-dimensional QKD over standard optical fiber networks for their ease of generation and manipulation, as well as for their resilience over fiber transmission. While energy-time entanglement has been widely exploited for high-dimensional QKD—through, e.g., large alphabets[20,21] and dispersive optics[22,23] protocols—time-bin entangled qudits are still untapped for quantum communications, despite their potential to enhance a variety of protocols such as quantum state teleportation[24,25] and superdense coding[26]. These limits are mostly due to the current interferometric schemes used to prepare and process multiple time bins. Increasing the number of arms in fiber-based interferometers leads to phase instability, timing inaccuracy, and low system efficiency (e.g., additional 3 dB splitting loss each time the number of time modes doubles[27,28]). These drawbacks inevitably affect the processing speeds and transmission rates achievable with time-bin qudits, which are far from those typical of standard telecommunications. Solutions realized so far to address these issues have been limited to prepare-and-measure QKD implementations[29–31], often relying on the use of free-space interferometers[30], which hamper the scalability of platforms towards higher dimensions. Approaches towards scalability have been proposed, for instance, by recurring to hybrid time-path encoding in multicore fibers, in which a 2-level path encoding is added to an existing 2-level time bin to enable an effective 4-level prepare-and-measure QKD implementation[13]. Such a hybrid approach further enabled to mitigate a main drawback in certain time-bin encoding schemes, i.e., low effective qudit rates. Specifically, in systems where the time-bin spacing is the same as the period of the pulse repetition, increasing the number of time bins per photon state, and hence the number of bits, comes at the price of reducing the repetition rate[31].

Here, we demonstrate a modular, fiber-pigtailed, integrated photonic platform enabling the generation and processing of picosecond-spaced time-bin entangled qudits up to $d = 8$ levels in the telecommunication C band, via an on-chip interferometric cascade (OIC). We utilize this framework to experimentally demonstrate the Bennett-Brassard-Mermin 1992 (BBM92) protocol[2] with time-bin entangled qudits (specifically, $d = 4$ levels or ququarts). To further show the capability of our platform for long-distance quantum communications, we extended this QKD experiment over a 60 km-long standard telecommunication fiber link. The generation of entangled qudits with a time-bin spacing in the picosecond regime, as enabled by the OIC, makes it possible to operate the entangled qudits at processing speeds typical of standard telecommunication systems (10 s of GBaud, where Baud defines the processable symbols per second or modulation rate across a communication channel[32]), as well as to increase the number of time bins per photon state without reducing the repetition rate of the system. We show an effective qudit rate increase by comparing the secret key rates achieved with entangled ququarts with those obtained with entangled qubits for the same experimental conditions. Our reconfigurable photonic framework further shows the advantage of flexibly increasing the Hilbert space (i.e., the quantum information capacity) of the entangled qudits within a low time-bandwidth product[30] (TBP, i.e., the dimensionless area occupied by the entangled qudits in the time and frequency domains), thus allowing the efficient usage of time-frequency space in terms of quantum state transmission and processing.

## Results

### Approach for entangled qudit generation and processing

The integrated platform used for the generation and processing of time-bin entangled qudits consists of a low-loss, programmable OIC[33,34] and a 45 cm-long on-chip spiral waveguide[35] (see Fig. 1a and "Methods"). Integrated photonic waveguides enable the scalable generation of high-dimensional entanglement[36], as well as they benefit from advances in lithographic fabrication techniques, which lead to short optical paths, low phase drifts, and delay times of a few picoseconds[37–39]. The OIC comprises an electronically switchable sequence of multiple unbalanced Mach-Zehnder interferometers (MZIs) in a waveguide-connected cascade (see Fig. 1a and Supplementary Note 1.2). We exploit the OIC as a multi-path optical splitter to prepare a pump pulse sequence (at a repetition rate of 250 MHz, corresponding to a 4 ns delay between pump burst sequences, see "Methods") for the generation of time-bin entangled qudits. By activating selected MZIs, an input laser pulse can be split into a coherent train of $d$ pulses with discrete spacing and equal amplitudes. It is worth noting that the OIC has a fully path-connected cascade of MZIs, which ensures a constant device loss for any chosen dimensionality. The prepared train goes through optical amplification, ultrafast (>10 GHz) phase modulation, and spectral filtering before being launched into the spiral waveguide. Here, spontaneous four-wave mixing takes place, in turn generating $d$-level time-bin entangled photon pairs (signal: $s$, and idler: $i$) within the telecommunications C band. The generated states are of the form[40]

$$|\Psi\rangle_d = \frac{1}{\sqrt{d}} \sum_{k=0}^{d-1} e^{i\theta_k} |k\rangle_s |k\rangle_i, \qquad (1)$$

where $k$ denotes the time bins and $e^{i\theta_k}$ describes the relative phases between the prepared pulses. Using this scheme, we generated 4- and 8-level entangled photon pairs featuring a time bin spacing of 64 ps and 32 ps, respectively, which are then spectrally filtered over a 200 GHz bandwidth through a commercially available dense wavelength division multiplexer (DWDM). The picosecond spacing of the spectrally broadband time bins within a prescribed communication frequency channel allows the proposed entangled qudits to reach high quantum information density, which can be defined as $d^N/(\Delta t \times \Delta\nu)$. In this expression, $d^N$ denotes the Hilbert space dimensionality (with $d$ and $N$ being the number of levels and photons, respectively), and $\Delta t \times \Delta\nu$ is the TBP of the entire $N$-photon state (with $\Delta t$ and $\Delta\nu$ being the time-bin width and the bandwidth, respectively). Specifically, we achieved a quantum information density of $4^2/(0.064 \text{ ns} \times 4 \times 200 \text{ GHz} \times 2) \approx 0.156$ for a 4-level state, and $8^2/(0.032 \text{ ns} \times 8 \times 200 \text{ GHz} \times 2) = 0.625$ for an 8-level state. As a comparison, for example, microring resonators generating frequency combs of 200 GHz spacing and full-time width of a few nanoseconds[18,19,40,41] yield a quantum information density as low as 0.004 for the 4-level state[40] and 0.009 for the 8-level state[41]. We note that a similar quantum information density of 0.148 has recently been achieved for 4-level states, by leveraging a frequency comb spacing as narrow as 15 GHz (yet, within hundreds of ps-large temporal width[42]), see Fig. 1b,c for further details[15,18,19,27,40–44].

The same OIC was utilized to analyze the entangled qudits, in particular for quantum interference and quantum state tomography (QST)[45,46], which were implemented by accessing the individual time

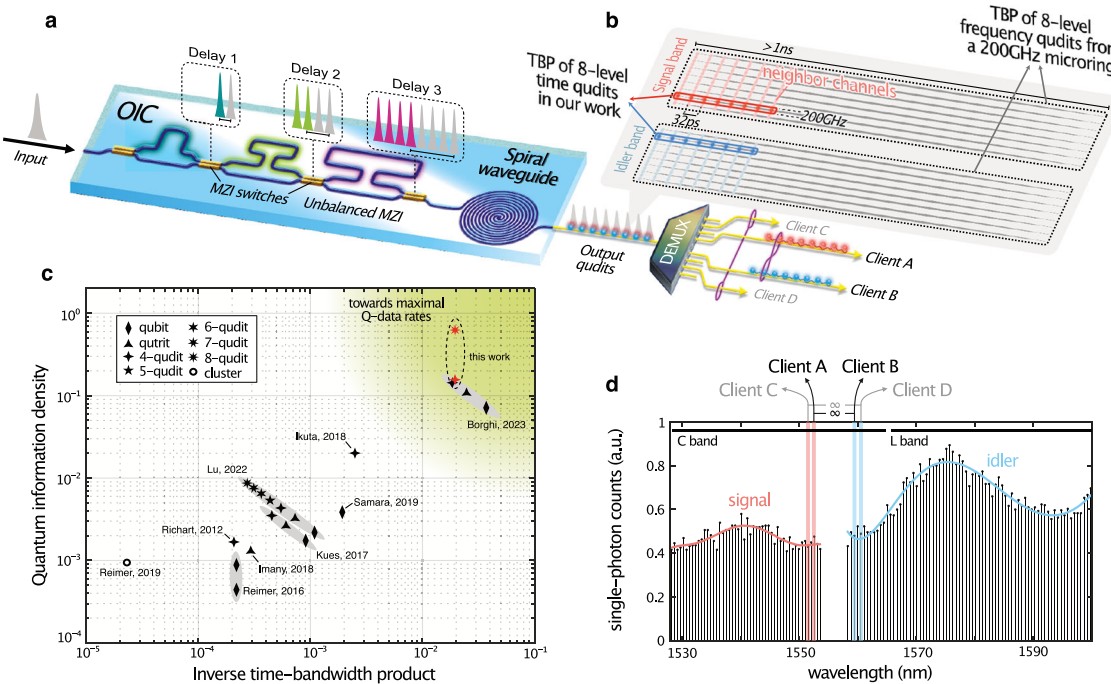

**Fig. 1 | On-chip generation of time-bin entangled qudits. a** Scheme for the generation of $d$-level time-bin entangled photon states, based on a switchable cascade of unbalanced Mach-Zehnder interferometers (MZIs) followed by a spiral waveguide. MZIs were used to split the input pump into a train of $d$ pulses, the temporal separation of which is given by the different interferometric path lengths. The burst then excited the spiral waveguide in such a way to generate signal and idler photonic qudits via spontaneous four-wave mixing. These high-dimensional entangled qudits were individually distributed to multiple users by different frequency channels through a demultiplexing (DEMUX) scheme. **b** Comparison between the time-bandwidth product (TBP) attainable with our platform and a microring resonator in the case of an 8-level entangled photon state. The example refers to a microring with a free spectral range of 200 GHz generating photon pairs with a −30 dB time width of 2 ns, yielding a TBP as high as 6400. In stark contrast,

our platform generated 8-level entangled qudits with a 32 ps time mode spacing that, combined with 200 GHz filtering −30 dB bandwidth, gave rise to a TBP as low as 102.4. **c** Current state-of-art in scaling the photonic Hilbert space size per TBP of two-photon sources using time-bin and/or frequency-bin entanglement. A decrease can be noted as the photon state dimensionality increases for sources highlighted in gray, which is intrinsic to current approaches dealing with two-photon frequency combs. The full temporal width of the frequency modes generated from microring resonators is evaluated by their Q-factors and spectral line widths. The compactness of the time-bin entangled states enabled through our platform allows to bypass this limitation (red symbols). **d** Measured single-photon counts per 100 GHz channel over the C and L telecommunications bands. OIC: on-chip interferometer cascade. Adapted with permission from ref. 34 © Optica Publishing Group.

bins via temporal gating[47], and by superimposing them through coherent mixing (see "Methods" and Supplementary Note 1.1). We obtain arbitrary phased control over the quantum state by utilizing an external phase modulator driven by a programmable arbitrary waveform generator (AWG) before the OIC. This allowed us to coherently mix the time-bin qudits at ultrafast speeds without modifying the OIC configuration. To bypass the ~52 ps-jitter time limitation of our superconducting nanowire single-photon detectors (SNSPDs), we utilized external ultrashort (25 ps) temporal gating[47] of the signal and idler photons, thus suppressing photon counts from neighboring time bins (see Fig. 2b, c). We note that enlarging the spacing of the time bins via an active optical nonlinear scheme[30] or a time lens scheme[48] is a possible way to enable ultrafast detection. However, this would come at the cost of a larger TBP, which would increase the time-domain resources needed over the communication channel.

### Characterization of 4-level time-bin entanglement
To generate and process the time-bin entangled ququarts ($d = 4$), we configured the OIC to activate two cascaded MZIs, consequently producing a train of four pump pulses with 64 ps spacing. This burst was used to pump the spiral waveguide and generate a photon state in the form $|\Psi\rangle = \frac{1}{2}\sum_{k=0}^{3} e^{i\theta_k}|k\rangle_s|k\rangle_i$ (see Fig. 2a). To analyze 4-level quantum interference, we applied a temporal gating by selecting the central time bin out of superposition states of all the time bins from the OIC (see Fig. 2b and "Methods"). To show the potential of our platform for telecommunication applications, we selected two signal-idler pairs over the C band, specifically, channels H22-H30 (corresponding to

1559.39 nm and 1552.93 nm, respectively) and channels H20-H32 (corresponding to 1561.01 nm and 1551.32 nm, respectively), see Fig. 1d. As shown in Fig. 3a, b, we measured raw visibilities of 90.14% ± 1.08% and 89.45% ± 1.14% per signal-idler pair (channels H22-H30 and H20-H32, respectively). These increased to 97.89% ± 1.40% (for H22-H30) and 96.03% ± 1.34% (for H20-H32) after background noise subtraction. All values exceeded the threshold of 81.70% needed to violate the CGLMP (Collins-Gisin-Linden-Massar-Popescu) inequality for $d = 4$ levels[28,49,50], which provides a (partially) device-independent certification of entanglement while demonstrating, at the same time, the reliability of our device[50].

Next, to measure the QST of the time-bin entangled ququarts[51], we used a complete set of quantum state projections that were experimentally accessible with our photonic chip, allowing us to perform 144 measurements only for quantum state density matrix reconstruction, while still obtaining all 256 combinations typically required for full QST (see Methods). We used the retrieved density matrix to verify its entanglement of formation[52,53] $E_{OF}(\rho)$, which resulted to be 1.992, meaning that the entangled photonic qudits carried almost 2 *ebit* of entanglement (the bound for a 4-level quantum system[52] is 2, see "Methods"). From QST measurement, we further extracted a quantum state fidelity $F = 84.44\% \pm 1.03\%$ (see Fig. 3c) via maximal likelihood estimation[45,46,54].

### Qudit scaling to 8-level time-bin entanglement
To show the scalability of our photonic platform to higher dimensions, we generated and processed 8-level entangled qudits as well. We

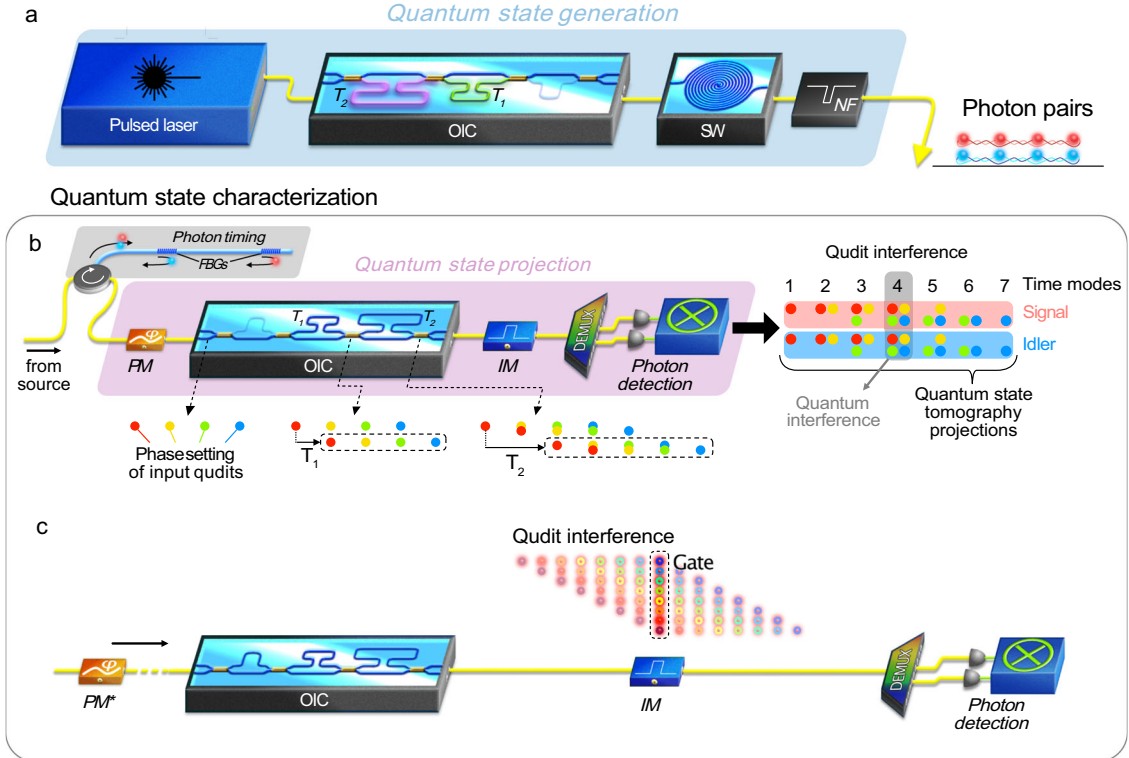

**Fig. 2 | Schematics of quantum state generation and processing for 4- and 8-level entangled qudits. a** Simplified experimental setup for the generation of picosecond-spaced high-dimensional time-bin entangled photonic qudits. **b** Operation principle for 4-level quantum state processing using the OIC and external phase modulation (see "Methods" and Supplementary Note 1). **c** Simplified scheme for 8-level quantum interference measurements, where the phase of each photon time bin was prepared by applying a phase modulation to the pump before photon generation. For clarity, the position of the phase modulator (PM*) is indicated in the setup. We applied temporal gating to separate the central interference modes and hence account for the fact that the time bin spacing (i.e., 32 ps and 64 ps) approaches the jitter time (i.e., ~52 ps) of the superconducting nanowire single-photon detectors. OIC: on-chip interferometer cascade, SW: spiral waveguide, NF: notch filter, FBG: fiber Bragg grating, PM: phase modulator, IM: intensity modulator, DEMUX: demultiplexer. Adapted with permission from ref. 34 © Optica Publishing Group.

activated three cascaded MZIs to produce a train of eight 32 ps-spaced pump pulses. This burst was used to pump the spiral waveguide, yielding the generation of an entangled photon state in the form $|\psi\rangle = \frac{1}{2\sqrt{2}}\sum_{k=0}^{7}e^{i\theta_k}|k\rangle_s|k\rangle_i$. By adopting the same temporal gating[47] technique used for the 4-level situation, we retrieved the joint temporal distribution[55], revealing the correlations between the eight different time bins (see Fig. 3d). We measured quantum interference by gating the central time bin out of the fifteen superposition events from the OIC (see Fig. 2c). For channels H22-H30 and H20-H32, we extracted raw visibilities of 91.23% ± 1.04% and 90.76% ± 1.01%, which, after background noise subtraction, became 98.80% ± 1.22% and 99.26% ± 1.29%, respectively (see Fig. 3e, f). In both cases, we exceeded the threshold of 89.56% necessary to violate the CGLMP inequality for $d = 8$ levels[28] (see "Methods").

## BBM92 QKD protocol with entangled qudits

To prove the suitability of our photonic platform for QKD applications, as well as an increase in the effective bit rate capacity, we implemented a BBM92-like scheme extended to ququarts. Benefitting from the picosecond spacing of the time-bin qudits enabled by the OIC, we could implement the QKD protocol by further increasing the repetition rate of the laser to 1 GHz, corresponding to a 1-ns delay between pump burst sequences (see "Methods"). Specifically, we implemented a BBM92-like protocol in which two clients—Alice and Bob—each receive a photon from a photon pair source[2,5]. Alice and Bob randomly choose a basis that is selected from two mutually unbiased bases[56] (the computational and phase bases), in which their respective photon will be measured. Alice and Bob then use the measurement outcomes to establish the secret key and identify any eavesdropping action by using a sifting procedure to recover the so-called "sifted" key (see "Methods"). In our work, the computational and the phase bases are given by the individual time bins (here called time basis) and their superposition, respectively[29]. We assessed the security of the QKD scheme by evaluating the quantum bit error rate (QBER)[12,57]. Specifically, we measured the QBER in both the time and the phase bases (see Fig. 4a), where 4% of the events were utilized for the time basis, while all events were used for the phase basis to compensate for the higher losses due to phase modulation (see Methods). Our QKD experiment ran for five hours, yielding an average QBER of 10.97% and 13.05% for the time and the phase basis, respectively. The obtained QBERs are well within the threshold of 18.93% tolerated by standard BB84 protocols based on ququarts[2] (see Fig. 4b and "Methods"). Over the same time lapse, we measured an average secret key rate of 2.04 kbit/s, which was determined through finite-key analysis (see Fig. 4c, "Methods" and Supplementary Note 2.4). The QKD scheme presented in our work can potentially enable secret key rates of 102.8 kbit/s by simply replacing the SNSPDs with state-of-the-art detectors[58] featuring lower jitter times (e.g., <30 ps), see Fig. 4d and Supplementary Note 2.2 and Note 4.

Finally, we extended the QKD experiment over a 60 km long telecommunication fiber link (see Supplementary Note 2.5). From this, we measured quantum interference after entangled state transmission yielding a raw visibility of 89.34% ± 2.55% and a secret key rate of 37 bits/s (potentially increasable to 1.93 kbit/s when suppressing the losses of the external temporal gating system). To demonstrate the enhancement in the effective key rates achievable via 4-level entangled qudits with respect to their 2-level counterparts, we reproduced the

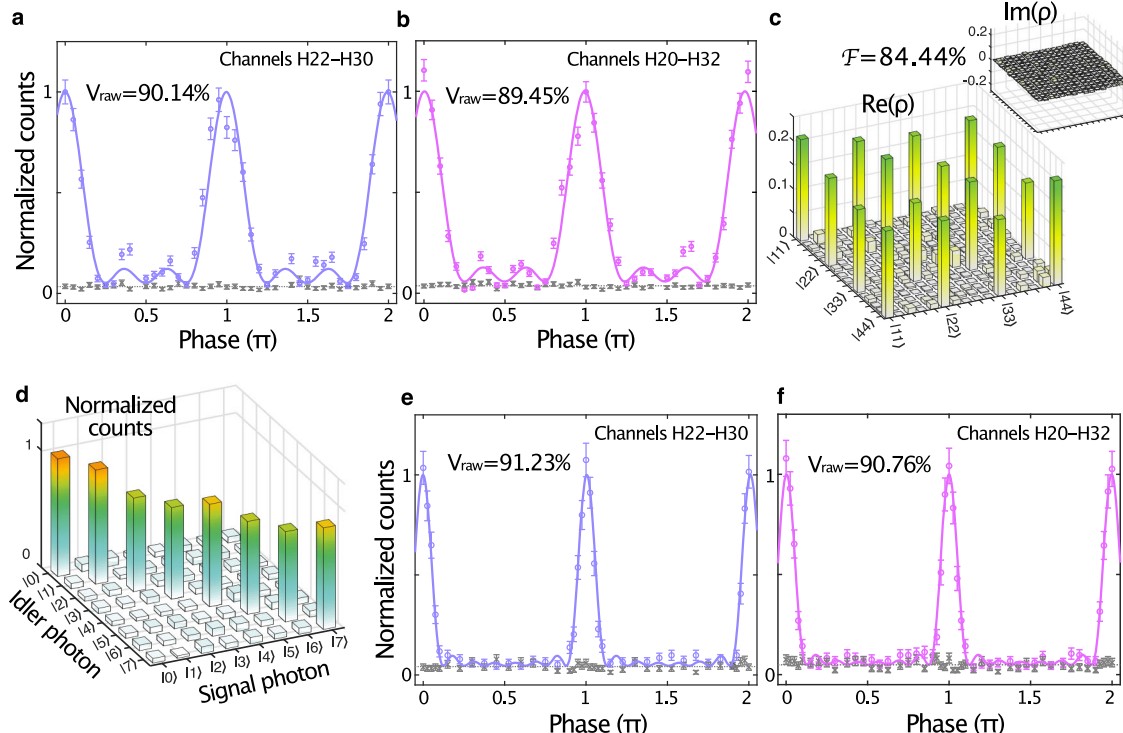

**Fig. 3 | Quantum interference and quantum state tomography for 4- and 8-level entangled qudits. a, b** Measured two-photon quantum interferences of 4-level entangled qudits over the H22-H30 and H20-H32 channels, respectively. Extracted raw visibilities of 90.14% ± 1.08% (**a**) and 89.45% ± 1.14% (**b**) exceeded the threshold necessary to violate Bell's inequality for 4 levels (i.e., 81.70%). The gray markers correspond to accidental coincidences, which were used to evaluate the visibilities without background noise. **c** Density matrix ρ retrieved from quantum state tomography measurements performed in a system-tailored set of projections (see Methods), from which we extracted a fidelity of 84.44% ± 1.03% and an entanglement of formation equal to 1.992. **d** Measured joint temporal distribution showing correlations between the eight different time bins. **e, f** Measured two-photon quantum interferences of 8-level entangled qudits over the H22-H30 and H20-H32 channels, respectively. Extracted raw visibilities were 91.23% ± 1.04% (**e**) and 90.76% ± 1.01% (**f**), both exceeding the threshold necessary to violate Bell's inequality for 8 levels (i.e., 89.56%). The gray markers correspond to accidental coincidences, which were used to evaluate the visibilities without background noise. Error bars are estimated using Poisson statistics.

BBM92 protocol with entangled qubits (see Supplementary Note 2.5). Results showcasing such an enhancement are illustrated in Fig. 4d.

## Discussion

We have demonstrated an application-ready, low-loss, hybrid chip-to-fiber architecture capable of generating and processing picosecond-spaced time-bin entangled photonic qudits. We showed the suitability of the platform for quantum communications by implementing a BBM92-like QKD protocol with entangled qudits over the equivalent of a 60 km-long optical fiber link. This result, together with the use of multiple channels in the telecommunication C band, demonstrates the potential of our platform to address multiple clients on a metropolitan scale. Furthermore, we establish a dimensionality as high as 8 × 8 in the generation and processing of discrete time-bin entanglement. The system design of the OIC can be in principle scaled to higher time bins, for instance, by activating more unbalanced interferometers inside the on-chip device. The maximum accessible number of time bins in this platform is 256, which can potentially enable, alongside with using a sub-ps pulsed pumping scheme, the generation of 256-level time-bin entangled photonic qudits with a time bin spacing of 1 ps. The OIC hence allows for scaling up the qudit dimensionality to $2^N$ time bins ($N \leq 8$) under the fixed two-photon temporal width of 256 ps, thus keeping a potential maximum repetition rate of 3.9 GHz.

The secret key rates reported in this work still remain orders of magnitude below those demonstrated via, e.g., prepare-and-measure protocols, which can register real-time keys >100 Mbit/s[59]. An important factor that intrinsically limits the performance achievable with entangled photons is associated to the rate at which correlated photon pairs are probabilistically generated from the entanglement source. In our work, this limitation comes from the noise associated to the spiral waveguide used as entanglement source, which can be improved significantly by dedicated source engineering. Our spiral operates at a brightness of $1.018 \times 10^{-12}$ pairs/(mW²·GHz) per pulse and at a coincidence-to-accidental ratio (CAR) of 20. Noise from the source comprises undesirable spontaneous Raman scattering photons and multi-photon events (see Supplementary Note 3). Yet, despite the relatively low performance of our spiral waveguide (which could be improved, in the longer term, with ad hoc fabrication techniques), the effective key rate extracted from the QKD experiment is much higher when compared to entangled qubit demonstrations[6,60,61].

The main scope of this work is indeed to show that the proposed on-chip interferometric framework has the potential to compensate some of the limits associated with entanglement sources and enable high secret key rates also in entanglement-based QKD protocols. On-chip waveguide sources made from other materials offer higher brightness, less or restricted Raman gain, and thus CAR values that are orders of magnitudes higher than ours[62–64]. Such sources can improve our system by enabling photon pair generation rates in the order of a few MHz under significantly lower pump peak powers (a few mW). These conditions can also reduce uncorrelated and multi-photon events, ultimately improving the security of the QKD scheme. Furthermore, future OIC designs on electro-active material platforms, involving active phase controls on the unbalanced MZIs to fully process the entangled qudits within the OIC (i.e., without external phase modulation) will reduce the overall processing losses. Such designs

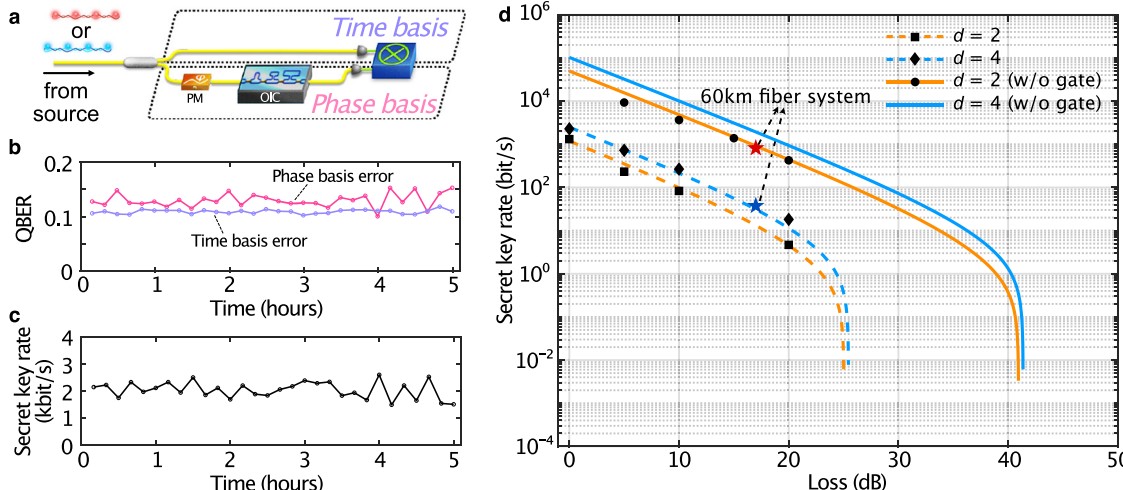

**Fig. 4 | BBM92-like scheme using time-bin entangled ququarts ($d = 4$) and comparison with qubits ($d = 2$). a** Simplified scheme showing the transmission of the signal photon to Alice's setup and the idler photon to Bob's setup for time and phase basis measurements. For simplicity, only one of the measurement setups is shown. The photons passed through a beam splitter to randomly experience either the time or the phase basis. For time basis measurements, we projected the qudits into one of the time bins by direct detection. For phase basis measurements, we projected the qudits into one of the phase vectors by using a phase modulator and the OIC. **b** QBER of time (phase) basis with an overall average error of $e_t = 10.97\%$ ($e_f = 13.05\%$). **c** Secret key rates measured over 5 h, showing an overall average key rate of 2.04 kbit/s. Each data point represents 10 minutes of acquisition time. **d** Secret key rates for (blue) 4-level and (orange) 2-level QKD versus channel loss (dB). The diamond and square markers show the experimental data acquired for entangled-ququart and entangled-qubit QKD schemes, respectively, when external temporal gating was applied. The circle markers represent the experimental data for entangled-qubit QKD without external temporal gating (w/o gate), i.e., without the loss from this system. This enhances the key rates by a factor of ~40, which matches the estimated efficiency of 2.51% for the external temporal gating system (see Supplementary Notes 2, 4). The blue and red star markers represent experimental data when 60 km of fiber (17 dB loss) is added to the system. The dashed and solid lines show simulation results. PM: phase modulator, OIC: on-chip interferometer cascade.

will allow for more effective generated keys, particularly for the phase basis (see Supplementary Note 1.1).

We note that BBM92 protocols based on simultaneous entanglement (hyperentanglement) in the time bins and in the polarization modes have been recently demonstrated[21]. However, while those schemes give access to a high-dimensional Hilbert space, the use of polarization limits the protocol to a qubit encoding. The BBM92-like QKD scheme demonstrated in our work shows the direct generation of the entangled photons from the source and not during the quantum state processing stage. Our results provide the tools to allow the manipulation and processing of time-bin entangled photonic qudits at standard telecommunication speeds (>16 GBaud), as well as to maximize the quantum information capacity per single communication channel by minimizing the TBP of the entangled qudits.

While the jitter time of our single-photon detectors was an obstacle to resolving the picosecond-spaced qudits, with the advent of state-of-the-art SNSPDs featuring picosecond-level jitter times[58,65,66], it will become possible to resolve narrow-spaced time bins. More in general, a variety of quantum communication protocols based on time-entanglement distribution will benefit from reducing the jitter times of the detectors. Protocols based on time-entanglement, both in its continuous and discrete forms, are constrained by the large jitter times of single-photon detectors when aiming to maximize the bit rate for high-speed implementations. For instance, energy-time entanglement schemes[20–22], where the photons' times-of-arrival are defined via discretization into time frames and bins during quantum state processing, necessitates the optimization of both time frames and bins to minimize error rates due to jitter times. Yet, we note that this is not the case for time-bin entanglement implementations[17–19], where the photons' times-of-arrival are predefined already at the entanglement source. In these schemes, error rates can be minimized through temporal post-selection, as done in our work.

The advancement in reducing jitter and dead times in single-photon detectors will lead to an increase in the dimensionality and bit rates achievable via time-bin qudits for ultrafast implementations

based on single- and entangled photons. If combined with on-chip interferometric systems as the one presented in our work, resolving narrow-spaced entangled time bins can ultimately lead to an enhancement of the bit capacity within a predefined temporal width of the photon states, without sacrificing the repetition rate of the system. Our work further enables the telecom-ready usage of quantum information processing and demonstrates the significant potential of time-bin entangled photonic systems to achieve high data rates and security levels for a variety of quantum communication protocols in optical fiber links over long distances.

## Methods
### Experimental setup
The experimental setup utilized for entangled photon generation and processing is reported in Supplementary Fig. 1 of the Supplementary Information. The OIC and a 45 cm-long spiral waveguide were fabricated on a single CMOS-compatible photonic chip made of high-index doped silica[35]. The fiber-pigtailed OIC featured an overall input-to-output loss of <4.5 dB. The $d$-fold pump pulse train was prepared from a mode-locked laser (Menlo Systems Inc., FC1500-250-WG, 250 MHz repetition rate), which was spectrally filtered to ~5 ps pulse duration centered at 1556.15 nm (corresponding to the H26 telecom channel). The generated pulses were amplified with an erbium-doped fiber amplifier (EDFA, Keopsys, CEFA-C-BO-HP) and then launched into the spiral waveguide to induce spontaneous four-wave-mixing (SFWM) for the generation of signal and idler photon pairs. Photons were then directed into a programmable filter (Finisar WaveShaper 4000 A), an electro-optic phase modulator (EO-Space, PM-5VEK-40-PFA-PFA-UV-UL), and then back into the OIC for quantum state processing.

After coherent mixing of the time bins in the OIC, the signal and idler photons were separated into two fiber channels using a standard wavelength-division multiplexer (LightWave2020). After demultiplexing, we used two 40 GHz intensity modulators driven by a 62.5 GSa/s AWG (Keysight) to implement 25 ps-wide of temporal gating[47] for the signal and idler photons. The photons were then directed into two

SNSPDs (Quantum Opus, 85% photon efficiency at 1550 nm, ~52 ps jitter time, and 20 ns dead time). We measured photon coincidences by using a time-tagging electronic unit (PicoQuant HydraHarp).

We note that the repetition rate of the mode-locked laser was increased to 1 GHz for the QKD experiment. To this end, we made use of a system comprising three polarization maintaining (PM) beam splitters (BSs) and short PM fibers of different lengths. Between the first and the second BSs, the length difference of the two fibers is 2 ns, thus doubling the repetition rate to 500 MHz. Then, between the second and the third BS, the length difference between the fibers is 1 ns, thus doubling the repetition rate from 500 MHz to 1 GHz.

### Quantum interference measurements

Coherent mixing of the time bins was achieved by propagating the entangled states through the OIC, where the modes experienced splitting, delaying, and partial recombination (see Fig. 2b). Arbitrary phase modulation of each time bin was externally achieved through a 40 GHz phase modulator driven by a 62.5 GSa/s AWG (Keysight). We realized projection measurements in the form[40]

$$|\psi_{\text{proj}}\rangle_{s,i} = \frac{1}{d}\left(\sum_{k_s=0}^{d-1} e^{ik_s\theta}|k_s\rangle\right)_s \left(\sum_{k_i=0}^{d-1} e^{ik_i\theta}|k_i\rangle\right)_i, \quad (2)$$

where $k_s$ and $k_i$ denote the time bin of the signal and the idler photon, respectively, and $\theta$ is the relative phase difference between the neighboring time bins.

### Quantum state tomography and entanglement of formation for time-bin entangled ququarts

To perform 4-level QST, we used a frequency-to-time mapping scheme[19] realized with a custom-made fiber Bragg grating (FBG) array, featuring an insertion loss of <1.5 dB (see Fig. 2b and Supplementary Fig. 1 of the Supplementary Information). This array consisted of two femtosecond-written FBGs[67] matched to the photon wavelengths, i.e., 1559.39 nm (signal) and 1552.93 nm (idler). The FBGs also separated the photons in time for individual phase modulation. Full QST necessitates 256 measurements resulting from the combination of a set of 16 single-photon projections, which typically require the mixing of three neighboring modes. The design of our OIC allowed us to perform all mode mixing except for three-mode mixing, due to a missing active phase control on the MZIs (see Supplementary Note 1.2). To address this issue, we considered a problem-specific complete set of 16 single-photon projections, which allowed us to perform full QST without three-mode mixing. We used the projections reported in Table 1. From QST, we further estimated the entanglement of formation, a monotone

### Table 1 | Projection measurements used for quantum state tomography

| | |
|---|---|
| $\|0\rangle$ | $\frac{1}{2}(\|0\rangle + \|1\rangle + \|2\rangle + i\|3\rangle)$ |
| $\frac{1}{\sqrt{2}}(\|0\rangle + \|1\rangle)$ | $\frac{1}{2}(\|0\rangle + i\|1\rangle - i\|2\rangle - \|3\rangle)$ |
| $\frac{1}{\sqrt{2}}(\|2\rangle + \|3\rangle)$ | $\frac{1}{2}(i\|0\rangle - i\|1\rangle + \|2\rangle - \|3\rangle)$ |
| $\|3\rangle$ | $\frac{1}{2}(-\|0\rangle + \|1\rangle - i\|2\rangle + \|3\rangle)$ |
| $\frac{1}{\sqrt{2}}(\|0\rangle - i\|1\rangle)$ | $\frac{1}{2}(-i\|0\rangle + i\|1\rangle - i\|2\rangle - \|3\rangle)$ |
| $\frac{1}{\sqrt{2}}(\|2\rangle + i\|3\rangle)$ | $\frac{1}{2}(\|0\rangle - i\|1\rangle + i\|2\rangle + \|3\rangle)$ |
| $\frac{1}{2}(\|0\rangle - \|1\rangle + \|2\rangle - \|3\rangle)$ | $\frac{1}{2}(\|0\rangle - i\|1\rangle - \|2\rangle + i\|3\rangle)$ |
| $\frac{1}{2}(\|0\rangle - i\|1\rangle - \|2\rangle - i\|3\rangle)$ | $\frac{1}{2}(\|0\rangle + \|1\rangle - i\|2\rangle - i\|3\rangle)$ |

Complete set of 16 single-photon projections used in our experiment to perform full QST without recurring to three-mode mixing.

used here to quantify the entanglement of the density matrix $\rho$ of our quart photon state. The entanglement of formation is defined as[52]

$$E_{OF} = -Tr(\rho_s \log_2 \rho_s), \quad (3)$$

where $\rho_s$ is the reduced density matrix of the signal photon, which is obtained from the density matrix of the whole quantum system (i.e., the photon pair) by tracing out the idler photon, i.e., $\rho_s = Tr_i(\rho)$. In the case of $d \times d$ maximally-entangled quantum systems (i.e., qudit Bell's states), the bound for the entanglement of formation is given by $E_{OF} \leq \log_2 d$ that, for $d = 4$, results in $\log_2 4 = 2$. We measured a value of 1.992, which certifies that the generated qudit states (i) carry almost 2 *ebit* for the entanglement of formation (i.e., they contain an amount of entanglement equivalent to almost two maximally entangled two-qudit pairs), and (ii) contain entanglement in (at least) $4 \times 4$ dimensions[52]. An *ebit* is defined as one unit of two-partite entanglement, i.e., the amount of entanglement that is contained in a maximally-entangled two-partite state (a Bell state).

### Derivation of the CGLMP (Collins-Gisin-Linden-Massar-Popescu) inequality threshold for $d = 8$ levels

The visibility threshold to verify $d$-level two-partite entanglement can be derived from the linear noise model. Here, the density matrix $\rho = |\psi\rangle\langle\psi|$ of a photon state $|\psi\rangle$, when affected by white noise, is modified as[68]

$$\rho' = (1 - \lambda_d)|\psi\rangle\langle\psi| + \lambda_d \frac{\mathbb{I}}{d^2}, \quad (4)$$

where $\lambda_d$ is the probability that the quantum state is affected by noise and $\mathbb{I}$ is the identity matrix. An 8-level entangled state can tolerate a critical noise mixture of $\lambda_{8,\text{thr}} = 0.318$ prior to losing entanglement. From this value, it is possible to derive the visibility threshold through the expression

$$V_d = \frac{d \cdot (1 - \lambda_d)}{2 + (1 - \lambda_d) \cdot (d - 2)} \quad (5)$$

which, for the 8-level case, gives rise to $V_8 = 89.56\%$.

### BBM92-like protocol with time-bin entangled photonic ququarts

QKD protocols based on high-dimensional photonic qudits rely on two sets of mutually unbiased bases[56] (i.e., the time and the phase bases) which are randomly chosen by Alice and Bob in order to perform measurements, in turn establishing secret keys[2,5]. The time basis is defined as $\{|k\rangle\}$ (with $k = 0, 1, 2, 3$), while the phase basis $\{|f_n\rangle\}$ reads

$$|f_n\rangle = \frac{1}{\sqrt{d}} \sum_{k=0}^{d-1} e^{i\frac{2\pi}{d}k \cdot n}|k\rangle, \quad (6)$$

(with $n = 0, 1, 2, 3$). When Alice and Bob randomly select the same basis, their measurements lead to correlated outcomes, which give rise to the sifted secret key. Uncorrelated data—resulting from measurements performed in different bases—are discarded in the sifting procedure. At the same time, Alice and Bob monitor the QBER. This method ensures that any attempt by an eavesdropper to measure the photon and thus intercept the key can be detected by the trusted users.

In our experiment, we randomly directed the photons to either the time- or the phase-measurement setup using a beam splitter-based delay scheme (see Supplementary Note 2.1). Such a system further allows photons to randomly arrive at each respective measurement setting either at the same time or with a temporal delay of 500 ps. In this way, we were able to measure multiple outcomes of the time and

the phase bases. Measurements in the time basis were achieved by applying external temporal gating to photons, which was accomplished by making use of the electro-optic intensity modulator driven by a 62.5 GSa/s AWG. Measurements in the phase basis were performed using electro-optic phase modulation followed by temporal gating. This setup had an additional transmission loss of ~17.5 dB, arising from the programmable filter (~5 dB), the phase modulator (~3 dB), the OIC (~7.5 dB), and the other fiber components (~2 dB). We note that future OIC designs could involve active, thermal phase controls on the unbalanced MZIs to ensure full quantum state processing within the OIC without external phase modulation. This would allow removing the electro-optic phase modulators, programmable filter, and some of the fiber components, thus reducing the total losses of the setup by ~9 dB (see Supplementary Note 1.1).

In the case of $d$-level systems, among all possible measurement outcomes, there is only one correct event and $d$-1 error events[69]. A random event corresponding to the correct outcome occurs with a probability of

$$P_{corr} = (1 - \lambda_d) + \frac{\lambda_d}{d}, \tag{7}$$

with $\lambda_d/d$ being the probability of an error outcome (i.e., $P_{err}$). From this, it is possible to derive the QBER as

$$e = (d-1)P_{err} = \frac{(d-1)\lambda_d}{d}. \tag{8}$$

This allows the QBER to be directly derived from quantum interference measurements and thus from the extracted visibilities. From 4-level quantum interference, we measured a visibility of $90.14\% \pm 1.08\%$ and thus a $\lambda_d = 17.95\%$ based on Eq. (5), from which we estimated a phase basis error of 13.46%, matching our QKD results. The secret key length $l$ was calculated through the expression[29,69]

$$l = \max_{\beta \in (0, \, \varepsilon_{sec}/4)} \left\lfloor n[\log_2 d - H_d(e_f + \delta)] - \text{Leak}_{EC} - \log_2 \frac{8}{\beta^4 \varepsilon_{cor}} \right\rfloor. \tag{9}$$

where $n$ is the sifted key rate, $e_t$ ($e_f$) is the measured QBER of the time (phase) basis, $\delta$ is the statistical noise due to finite-key statistics, $\text{Leak}_{EC}$ is the information leakages for error correction and $H_d(x)$ is the $d$-level Shannon entropy function: $H_d(x) = -x\log_2(x/(d-1)) - (1-x)\log_2(1-x)$. $\beta$ is a constrained optimization parameter and we set $\varepsilon_{sec} = 10^{-9}$ and $\varepsilon_{cor} = 10^{-12}$ (see Supplementary Note 2.4).

## Data availability
The data that support the findings of this study are available from the corresponding authors upon request.

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

## Acknowledgements

We thank QuantumOpus and Nick Bertone from OptoElectronics Components for their help and for providing us with state-of-the-art superconducting nanowire single-photon detection equipment. S.S. and N.M. acknowledge the financial support from Mitacs Elevate Thematic Quantum Science. H.Y. acknowledges the financial support from the China Scholarship Council (CSC). M.C. acknowledges support from an NSERC (Natural Sciences and Engineering Council of Canada) Banting program and from an FRQNT (Fonds de recherche du Québec—Nature et technologies) PBEEE (Bourses d'excellence pour étudiants(es) étrangers(ères)) program. B.W. acknowledges the support of the Conseil Régional Nouvelle-Aquitaine (the SPINAL project). This work has received funding from the European Research Council (ERC) under the European Union's Horizon 2020 research and innovation program under grant agreement no. 950618 (the STREAMLINE project). T.G., R.K., and S.N. acknowledge financial support from Deutsche Forschungsgemeinschaft (259607349/GRK2101 and 455425131) as well as funding by BMBF within the funding program "Quantum Technologies—from basic research to market" (13N16028). W.J.M. acknowledges partial support through the Japanese JSPS KAKENHI Grant no. 21H04880. Z.W. acknowledges the National Key

Research and Development Program of China (2019YFB2203400) and the "111 Project" (B20030). DJM acknowledges funding support from the Australian Research Council Centre of Excellence COMBS (Centre for Optical Microcombs for Breakthrough Science), Project (No. CE230100006). R.M. acknowledges support from the Canada Research Chair program and NSERC through the following projects: AQUA ALLRP 587602-23, QuEnSi ALLRP 578468-22, Consortium on Integrated Quantum Photonics with Ferroelectric Materials ALLRP 587352-23, HyperSpace ALLRP 569583-21, and from FRQNT, through the project AdéQuATS FRQNT 328872.

## Author contributions

H.Y., M.C. and S.S. conceived the project. H.Y., S.S., M.C. and N.M. carried out the experiment. S.S., H.Y., and W.J.M. performed the theoretical analysis. H.Y., M.C. and S.S. analyzed the data. B.C. and J.A. developed the electronics part and implemented the external temporal gating. B.E.L., S.T.C., and D.J.M. designed and fabricated the photonic chips. M.C., B.F., R.H. and B.W. developed the package and electronic control of the photonic chip. R.G.K., T.A.G., and S.N. designed, fabricated, and characterized the custom fiber Bragg grating. R.M. and Z.W. supervised the project. All authors reviewed and edited the manuscript.

## Competing interests

S.S., N.M., R.H., and R.M. declare that a provisional patent application based on this work has been filed. The remaining authors declare no competing interests.
