## [Transparent Peer Review file · Nature Communications]

Quantum key distribution implemented with d-level time-bin entangled photons

Corresponding Author: Dr Stefania Sciara

Version 0:

Reviewer comments:

Reviewer #1

(Remarks to the Author)

In the work "Quantum key distribution implemented with d-level time-bin entangled photon" the authors present an entanglement based QKD experiment using a modified BBM92 protocol for a four-dimensional space.

The work is in my opinion correct and well presented. However, I have some doubts I would like the authors to address before I could express myself over the suitability for publication on the journal Nature Communication.

I think that my main concern is related to the multi-dimensional aspect of the protocol. In fact, in the example given in previous works that uses qudits in time-bin based, prepared and measured protocol, the increased number of bits generated per symbol goes to the expense of the repetition rate of the system. And while the number of time bin uses scales the number of bits generated in a logarithmic way, the expense on the rate is linear, making the protocol not useful after with a qudit higher than dimension 4. This last remark is true when the separation of two symbols is the same as the separation of two time-bins (the modulation is as fast as possible), which is not the case here, since the symbols are much further apart (4 ns, 250 MHz repetition rate) than the time bins (64 ps for the 4-levels system, 32 ps for the 8-levels system). However, I would like for the authors to argue this issue a little bit better because if taken at face value, the approach to use higher dimensions in time-bin encoding schemes could be disregarded following my previous argument.

One of the advantages of high-dimensional QKD is the higher resilience to noise. As the author shows, the modified BBM92 protocol here proposed allows for up to 18% QBER with respect to the 11% of a standard (symmetric) BB84. However, the authors show a measured QBER of over 11% and a phase error of 13.46%. What is the main source of this noise? How much does it affect the performance of the protocol?

The authors show that the phase basis measurement has an additional 17.5 dB of loss, which is a quite severe mismatch with respect to the other basis. Did the author consider how this could affect the security of the protocol? And how does this affect performance? I would expect that the raw key rate, produced in the time-basis, is not affected. However, in the finite-key regime, having a very inefficient phase-basis measurement would force to unbalance the basis choice significantly, reducing consequently the raw key generation rate.

Reviewer #2

(Remarks to the Author)

The authors present a time-bin entanglement qudits generation based on an integrated photonic platform, achieving a high quantum information density. They also demonstrate an implementation of the BBM92 QKD protocol using the 4-level qudit entanglement. This work is impressive in its high technical quality. The multi-level entanglement generation method and the results are instructive, and its potential applications can attract broad interest in the field. I will consider its publication in a high impact journal, like Nature Communications, once the following comments are adequately addressed.

1. The power and repetition rate of the pump laser, as well as the brightness of the entanglement source, should be provided in the main text for a better understanding of the source.

2. Is the finite-key effect taken into consideration when calculating the experimental key rates? Note that this is an important part for QKD. Moreover, the reported QBER of 11.98% is much higher than standard BB84. What are the limiting factors on this high QBER? On this point, a recent review on the subject of QKD will be helpful for readers, e.g., [Reviews of Modern Physics 92, 025002 (2020)].

3. A recent work on frequency-bin qudit entanglement generation [Phys. Rev. Appl. 19, 064026 (2023)] has a frequency bin spacing of 15 GHz. The authors may add this reference to the manuscript and include it in the entanglement source comparison in Figure 1c.
4. Elaborate more clearly on the difference between the solid and dashed lines in Figure 4d, as well as the experimental setup corresponding to the blue and red stars. The data points for the 'ideal' case may be removed, because this is not the real data in your experiment. The solid lines are sufficient to show the expectation.
5. The authors report a key rate of 2.47 Kbps or potentially improve to 100.9 Kbps. But this is much lower than state-of-art real-time key rate of >110 Mbps [Nature Photonics 17, 416 (2023)]. One motivation of high-dimensional QKD is the potential of high secret rates. Unfortunately, this experiment does not prove this potential. Also, the reported rate here is not real time yet. The authors should carefully discuss this point and the application front of high-dimensional QKD on high key rates.
6. Throughout the manuscript, the timing jitter of the SNSPDs appears to be a main obstacle for characterizing the entangled qudits with an ultrashort time-bin spacing and further increasing the key rates in the implementation of BBM92 QKD. Please consider discussing how the performance of the SNSPDs will affect the qudit entanglement sources and their applications, which can be helpful for a broader readership.
7. Ref. [3] and [54] are the same.

Version 1:

Reviewer comments:

Reviewer #1

(Remarks to the Author)

First, I'd like to thank the authors for addressing my concerns with precision and thoroughness. I found their responses honest, and the corrections and additions to the paper appropriate.

After this revision, I can say that the technical aspects of the work, particularly those related to entanglement generation and integrated photonics, are incredibly interesting. The main weakness of the work, however, lies in the specific implementation of the protocol chosen by the authors.

As the authors also noted, entanglement-based QKD has several disadvantages compared to other QKD protocols. For example, entanglement-based QKD is generally less efficient than prepare-and-measure QKD, such as BB84-like protocols. Additionally, the configuration of entanglement-based protocols with a central source and two receivers at Alice and Bob introduces more vulnerabilities (with detectors being the most vulnerable component in this type of cryptography). It is often preferable to implement measurement-device-independent schemes, where an untrusted receiver station is in the middle, and Alice and Bob use simpler transmitters.

However, the authors have acknowledged these drawbacks in their responses and corresponding revisions with clarity and honesty.

In my opinion, this work represents a highly valuable technical contribution, where the photon-pair generation techniques used are benchmarked through a QKD protocol. Although the QKD results may be less significant in this context, they still lead me to recommend this work for publication in Nature Communications.

Reviewer #2

(Remarks to the Author)

The authors have address all my concerns. I am happy to recommend its publications as it is.

Response to Referees Letter – manuscript NCOMMS-24-24295

In this Letter of response to Referees, the reviewers' comments are reproduced in *italic font*, while our replies are highlighted in blue. The modifications made in the revised manuscript, including the added references, are highlighted in red.

Reviewer #1 (Remarks to the Author):

In the work “Quantum key distribution implemented with d -level time-bin entangled photon” the authors present an entanglement based QKD experiment using a modified BBM92 protocol for a four-dimensional space.

The work is in my opinion correct and well presented. However, I have some doubts I would like the authors to address before I could express myself over the suitability for publication on the journal Nature Communication.

We thank the Reviewer for acknowledging the correctness and presentation of our work. We thoroughly considered the doubts that were raised and hope we have addressed each concern in full.

***Q1.** I think that my main concern is related to the multi-dimensional aspect of the protocol. In fact, in the example given in previous works that uses qudits in time-bin based, prepared and measured protocol, the increased number of bits generated per symbol goes to the expense of the repetition rate of the system. And while the number of time bin uses scales the number of bits generated in a logarithmic way, the expense on the rate is linear, making the protocol not useful after with a qudit higher than dimension 4. This last remark is true when the when the separation of two symbols is the same as the separation of two time-bins (the modulation is as fast as possible), which is not the case here, since the symbols are much further apart (4 ns, 250 MHz repetition rate) than the time bins (64 ps for the 4-levels system, 32 ps for the 8-levels system). However, I would like for the authors to argue this issue a little bit better because if taken at face value, the approach to use higher dimensions in time-bin encoding schemes could be disregarded following my previous argument.*

A1. We thank the Reviewer for this valuable comment, which gives us the opportunity to further discuss the ability to generate narrow-spaced time bins through the on-chip interferometric system. We agree that the effective qudit rate is a drawback in certain time-bin encoding schemes, where increasing the number of time bins per symbol comes at the expense of reducing the repetition rate of the system. This is indeed due to the scaling mentioned by the Reviewer. Consequently, for a fixed number of time modes, qubit-wise detection allows for higher photon rates and correspondingly higher bit rates when compared to qudits. We acknowledge a very recent work from M. Zahidy *et al.* published on *Nature Communications* **15**, 651 (2024), cited as Ref. [17] of the revised manuscript, in which the authors tackle such a drawback by recurring to hybrid time-path encoding QKD in multicore fibers. They add a 2-level path encoding to the existing 2-level time-bin encoding, thus effectively reaching a 4-level implementation.

Yet, we still believe that our QKD scheme has the significant advantage of enabling an increase in the effective qudit rate without sacrificing the repetition rate of the system. Such a capability is made possible by our on-chip interferometric system, which delivers time-bin entangled two-photon states with a temporal width fixed at 256 ps for any 2^N dimensionality (N being the number of activated switches of the interferometer cascade). While quantum state characterization experiments were performed at a repetition rate of 250 MHz (4 ns), we implemented the QKD test by increasing the repetition rate to 1 GHz (1 ns) and by scaling up the number of time bins from 2 to 4, in order to corroborate this specific aspect. We could increase the repetition rate of the laser by making use of a system that comprised three polarization maintaining (PM) beam splitters (BSs) and short PM fibers of different lengths. Between the first and the second BSs, the length difference of the two fibers was 2 ns, thus doubling the repetition rate to 500 MHz. Then, between the second and the third BS, the length difference between the fibers was 1 ns, thus doubling the repetition rate from 500 MHz to 1 GHz.

Moreover, our on-chip interferometric system has the potential to maximize the repetition rate towards the limit of the overall two-photon temporal width of 256 ps is at the repetition rate of ~ 3.9 GHz (we do not have lasers operating at that rate, though).

To show an effective qudit rate enhancement, we compared the secret key rates achieved via entangled ququarts (64-ps spacing) with those achievable via qubits (128-ps spacing) at the same experimental conditions. As also illustrated in Fig. 4d of the main text (which we have revised and reported in this rebuttal letter, following Q4 from Reviewer #2), ultrafast entangled ququarts bring a significant key rate enhancement as compared to entangled qubits.

Beside the QKD experiment, we further demonstrated the capability of the on-chip interferometric system to scale of number of time bins (and, potentially, the bit capacity per photon state) up to 8 levels within a temporal width of 256 ps and at a fixed repetition rate of 250 MHz.

Based on the Reviewer's comment, we have carefully discussed these aspects in the revised manuscript.

We revised part of the abstract as follows:

...However, their use is hindered by phase instability, timing inaccuracy, and low scalability of interferometric schemes needed for time-bin processing⁴. **As well, increasing the number of time bins per photon state typically requires decreasing the repetition rate of the system, affecting in turn the effective qudit rates.** Here, we demonstrate a fiber-pigtailed, integrated photonic platform enabling the generation and processing of ultrafast time-bin entangled qudits in the telecommunication C band **via an on-chip interferometer system**. We implement the first experimental demonstration of the Bennett-Brassard-Mermin 1992⁵ quantum key distribution

protocol with time-bin entangled qudits and extend it over a 60-km-long optical fiber link, by showing dimensionality scaling without sacrificing the repetition rate.

We revised some parts of the introduction as follows:

Solutions realized so far to address these issues have been limited to prepare-and-measure QKD implementations^{3,31,32}, often relying on the use of free-space interferometers³¹, which hamper the scalability of the platforms towards higher dimensions. Excellent approaches towards scalability have been proposed, for instance, by recurring to hybrid time-path encoding in multicore fibers, in which a 2-level path encoding is added to an existing 2-level time bin to enable an effective 4-level prepare-and-measure QKD implementation¹⁷. Such a hybrid approach further enabled to mitigate a main drawback in certain time-bin encoding schemes, i.e., low effective qudit rates. Specifically, in systems where the time-bin spacing is the same as the period of the pulse repetition, increasing the number of time bins per photon state, and hence the number of bits, comes at the price of reducing the repetition rate³².

Here, we demonstrate a modular, fiber-pigtailed, integrated photonic platform enabling the generation and processing of ultrafast time-bin entangled qudits up to $d=8$ levels in the telecommunication C band, via an on-chip interferometric cascade (OIC).

The generation of the entangled qudits with a time-bin spacing in the picosecond regime, as enabled by the OIC, makes it possible to operate the entangled qudits at processing speeds typical of standard telecommunication systems (10s of GBaud, where Baud defines the processable symbols per second or modulation rate across a communication channel³³), as well as to increase the number of time bins per photon state without reducing the repetition rate of the system. We show an effective qudit rate increase by comparing the secret key rates achieved with entangled ququarts with those obtained with entangled qubits for the same experimental conditions.

In the introductory paragraph of the QKD experiment, we write:

To prove the suitability of our photonic platform for QKD applications, as well as an increase in the effective bit rate capacity, we implemented a BBM92-like scheme extended to ququarts. Benefitting from the ultrafast nature of the time-bin qudits enabled by the OIC, we could

implement the QKD protocol by further increasing the repetition rate of the laser to 1 GHz (corresponding to a 1-ns delay between pump burst sequences, see Methods). Specifically, we implemented a BBM92-like protocol in which two clients...

In the conclusion, we have added the following sentence:

The maximum accessible number of time bins in this platform is 256, which can potentially enable, alongside with using a sub-ps pulsed pumping scheme, the generation of 256-level time-bin entangled photonic qudits with a time bin spacing of 1 ps. The OIC hence allows for scaling up the qudit dimensionality to 2^N time bins ($N \leq 8$) under the fixed two-photon temporal width of 256 ps, thus keeping a potential maximum repetition rate of 3.9 GHz.

Furthermore, in the section dedicated to the description of the experimental setup, we have explained how we could increase the repetition rate of the laser from 250 MHz to 1 GHz. In this regard, we moved part of section S1.1 of the Supplementary Information to the Methods, in such a way to provide already in the main Article file a clearer description of the experimental setup.

The Methods section reads as follows:

Experimental setup. The experimental setup utilized for entangled photon generation and processing is reported in Fig. S1 of the Supplementary Information. The OIC and a 45cm-long spiral waveguide were fabricated on a single CMOS-compatible photonic chip made of high-index doped silica³⁶. The fiber-pigtailed OIC featured an overall input-to-output loss of < 4.5 dB. The d -fold pump pulse train was prepared from a mode-locked laser (Menlo Systems Inc., FC1500-250-WG, 250 MHz repetition rate), which was spectrally filtered to ~5 ps pulse duration centered at 1,556.15 nm (corresponding to the H26 telecom channel). The generated pulses were amplified with an erbium-doped fiber amplifier (EDFA, Keopsys, CEFA-C-BO-HP) and then launched into the spiral waveguide to induce spontaneous four-wave-mixing (SFWM) for the generation of signal and idler photon pairs. Photons were then directed into a programmable filter (Finisar WaveShaper 4000A), an electro-optic phase modulator (EO-Space, PM-5VEK-40-PFA-PFA-UV-UL), and then back into the OIC for quantum state processing.

After coherent mixing of the time bins in the OIC, the signal and idler photons were separated into two fiber channels using a standard wavelength-division multiplexer (LightWave2020). After demultiplexing, we used two 40 GHz intensity modulators driven by a 62.5 GSa/s AWG [arbitrary waveform generator] (Keysight) to implement 25ps-wide of temporal gating⁴⁸ for the signal and idler photons. The photons were then directed into two SNSPDs [superconducting nanowire single-photon detectors] (Quantum Opus, 85% photon efficiency at 1,550 nm, ~52 ps jitter time, and 20 ns dead time). We measured photon coincidences by using a time tagging electronic unit (PicoQuant HydraHarp).

We note that the repetition rate of the mode-locked laser was increased to 1 GHz for the QKD experiment. To this end, we made use of a system comprising three polarization maintaining (PM) beam splitters (BSs) and short PM fibers of different lengths. Between the first and the second BSs, the length difference of the two fibers is 2 ns, thus doubling the repetition rate to 500 MHz. Then, between the second and the third BS, the length difference between the fibers is 1ns, thus doubling the repetition rate from 500 MHz to 1 GHz.

The revised Section S1.1 of the Supplementary Information can be found at the end of A3 that we have provided to Q3 raised from the Reviewer.

Q2. One of the advantages of high-dimensional QKD is the higher resilience to noise. As the author shows, the modified BBM92 protocol here proposed allows for up to 18% QBER with respect to the 11% of a standard (symmetric) BB84. However, the authors show a measured QBER of over 11% and a phase error of 13.46%. What is the main source of this noise? How much does it affect the performance of the protocol?

A2. We thank the Reviewer for this comment. We carefully considered the questions about the main source of noise and the extent of which it affects the performance of the protocol. As the Reviewer pointed out, standard BB84 protocols based on qubit systems tolerate QBERs up to 11%, while QKD schemes based on 4-level qudits can increase such tolerance up to 18.93% (*Phys. Rev. Lett.* **88**, 127902 (2002) and *Phys. Rev. Appl.* **18**, 044027 (2022), respectively Refs. [16] and [67] of the revised manuscript). For this reason, in the original manuscript, we found it more reasonable to directly compare the measured QBER with the QBER associated to 4 levels (18.93%). Yet, we understand the Reviewer's concern. The main source of noise in the BBM92-like scheme proposed here comes from the spiral waveguide utilized for the generation of entangled photon pairs. The coincidence-to-accidental ratio (CAR) of the spiral waveguide is affected by unexpected spontaneous Raman scattering photons and multi-photon events.

To answer the Reviewer’s comment, we have added a new section in the Supplementary Information, named “Characterization of the spiral waveguide”, in which we provide a detailed theoretical and experimental study of the spiral waveguide, along with simulations to show its impact in our QKD implementation, including the 13.05% phase error (from further calculation made during the revision process). We also kindly ask the Reviewer to refer to A5 that we have provided to Reviewer#2.

The added section in the Supplementary Information is reported below.

S3. Characterization of the spiral waveguide

The signal and idler photon pairs generated from the spiral waveguide through SFWM follow a thermal distribution and their quantum state can be expressed as⁶

$$|\Psi\rangle = \sum_{n=0}^{+\infty} \left[\frac{\mu^n}{(\mu + 1)^{n+1}} \right]^{\frac{1}{2}} |n\rangle_s |n\rangle_i. \quad (\text{S.25})$$

Assuming a pulsed laser pumping scheme (as in our work), in this expression μ is the mean photon pair per pulse generated by the SFWM process, while n is the number of signal (s) and idler (i) photons generated per pulse. Assuming $\mu \ll 1$, the mean photon pair per pulse quadratically scales with the peak pump power P as $\mu(P) = aP^2$, where the coefficient a expresses the photon pair generation efficiency at a given pump peak power. In contrast, the generation of uncorrelated photons from Raman scattering scales linearly with the peak pump power. The probability of detecting any noise photons per pulse is given by $\xi(P) = bP + c$, where the coefficient b expresses the generation efficiency of Raman photons produced per pulse and c is a constant noise fraction that depends on the dark counts of the single-photon detectors. Assuming that the total collection efficiency η and the noise level ξ for signal and idler channel are the same, the two-photon coincidence probability results in

$$P_{co} = \eta^2 \mu(1 + \mu) + (\xi + \eta\mu)^2, \quad (\text{S.26})$$

with $P_{ac} = (\xi + \eta\mu)^2$ being the accidental probability, governed by uncorrelated Raman photons and multi-photon events.

Based on Eq. (S26), we can measure two-photon coincidences and accidentals at different pump powers, so as to estimate the coefficients a and b , and thus characterize the spiral waveguide. The mode-locked laser used to pump the spiral, centered at 1,556.15 nm and featuring a repetition rate of 250 MHz, was spectrally filtered to ~ 5 ps pulse duration. The total collection efficiency per

photon is $\eta = 8.8\%$, which includes the losses of the spiral waveguide (coupling losses, ~ 2.2 dB), of the programmable filter (~ 5 dB), of other fiber components (~ 2 dB), and of the SNSPDs (~ 0.7 dB). Fig. S5a shows the experimental results of the coincidence-to-accidental ratio (CAR) values at different peak pump powers, while Fig. S5b shows the CAR values versus the coincidence counts. By fitting the data points (respectively, solid red lines and black circles in Fig. S5) based on Eq. (S26), we obtain $a = 1.527 \times 10^{-4} \text{ W}^{-2}$ and $b = 1.722 \times 10^{-3} \text{ W}^{-1}$. The CAR could achieve a value of 50 for the low pump power regime, where the Raman noise is less detrimental. However, in our experiment, to balance the Raman noise and the photon pair generation rate from the SFWM process, we selected and operated at a CAR value of 20 for quantum interference and QKD implementations. This required pumping the spiral waveguide at a relatively high peak power of ~ 6.5 W. We note that the estimated generation efficiency of photon pairs produced by SFWM is orders of magnitude lower than for other on-chip waveguide sources⁷⁻⁹. The quality of the time-bin entangled qudits can be further improved by using on-chip two-photon sources with higher brightness and lower noise. The brightness of our spiral waveguide is 1.018×10^{-12} pairs/(mW²·GHz) per pulse. State-of-the-art on-chip waveguide sources made from other materials (e.g., silicon) report higher brightness and CAR values as compared to our spiral waveguide. For instance, in Ref.⁸ a brightness of 1.535×10^{-9} pairs/(mW²·GHz) per pulse and a CAR of 400 are achieved, while in Ref.⁷ the registered values amount to 2.993×10^{-5} pairs/(mW²·GHz) per pulse and a CAR of 1633, respectively. Such high efficiencies could enable \sim MHz levels of photon pair generation rates under significantly lower pump peak powers (a few mW), ultimately reducing Raman scattering effects and increasing the CAR. We note that, despite the relatively low performance of the spiral waveguide, our scheme could still achieve good results in terms of entangled qudits' quantum interference, quantum state tomography, and BBM92-like QKD protocol. This was possible due to the ability of leveraging the on-chip cascade interferometric system.

Based on the coefficients a and b obtained from the source characterization, we simulated the CAR values at different peak pump powers considering the total collection efficiencies associated to entangled ququart-based QKD measurements in the time and phase bases ($\eta_t = 1.50\%$ and $\eta_p = 0.026\%$, respectively). The results from simulations are shown in Fig. S6a. At a pump peak power of ~ 6.5 W, CAR values are still maintained at around 20 for both time and phase basis measurements (blue and red curves, respectively). In Fig. S6b we report the simulated QBER

versus the peak power for the time and phase bases (blue and red curves, respectively). Simulations match with the QBERs measured in our work for the time and phase bases (blue and red triangles, respectively). Finally, we calculated the estimated SKR by considering a 4-fold pump burst with a repetition rate of 1 GHz, based on the same collection efficiency and QBER shown before. The blue curve in Fig. S6c illustrates the simulated results, which indicate that, when the spiral waveguide is pumped with a 4-fold burst at ~ 6.5 W peak power, the SKR reaches a maximum of ~ 2.5 kbit/s. For our QKD demonstration, we chose the pump power based on these simulations. The black triangle in Fig. S6c shows the experimental output obtained for the SKR by using finite-key estimation, which results lower than the maximum obtained in the simulation curve.

Figure S5. Characterization of the spiral waveguide. **a)** CAR values at different peak pump powers. **b)** CAR values versus the coincidence counts. Black circles are experimental data points, while the two red curves are fitted based on Eq. (S26).

Figure S6. Simulations of the time and phase bases measurement according to the source characterization. **a)** CAR values at different peak pump powers considering the total collection efficiencies for the time ($\eta_t = 1.50\%$) and phase ($\eta_p = 0.026\%$) bases measurement. **b)** Simulated QBER versus the pump peak power for the time and phase bases. Blue and red triangles are experimental data. **c)** Estimated secret key rate. The black triangle represents the experimental data.

In addition, in the revised conclusion of the main text, we have added the following discussion:

The secret key rates reported in this work still remain orders of magnitude below those demonstrated via, e.g., prepare-and-measure protocols, which can register real-time keys >100 Mbit/s⁶⁰. An important factor that intrinsically limits the performance achievable with entangled photons is associated to the rate at which correlated photon pairs are probabilistically generated from the entanglement source. In our work, this limitation comes from the noise associated to the spiral waveguide used as entanglement source, which can be improved significantly by dedicated source engineering. Our spiral operates at a brightness of 1.018×10^{-12} pairs/(mW²·GHz) per pulse and at a coincidence-to-accidental ratio (CAR) of 20. Noise from the source comprises undesirable spontaneous Raman scattering photons and multi-photon events (see Supplementary Information S3). Yet, despite the relatively low performance of our spiral waveguide (which could be improved, in the longer term, with *ad hoc* fabrication techniques), the effective key rate extracted from the QKD experiment is significant when compared to entangled qubit demonstrations^{11,61,62}. The main scope of this work is indeed to show that the proposed on-chip interferometric framework has the potential to compensate some of the limits associated with entanglement sources and enable high secret key rates also in entanglement-based QKD protocols. On-chip waveguide sources made from other materials offer higher brightness, less or restricted Raman gain, and thus CAR values that are orders of magnitudes higher than ours^{63–65}. Such sources can improve our system by enabling photon pair generation rates in the order of a few MHz under significantly lower pump peak powers (a few mW). These conditions can also reduce uncorrelated and multi-photon events, ultimately improving the security of the QKD scheme. Furthermore, future OIC designs on electro-active material platforms, involving active phase controls on the unbalanced MZIs to fully process the entangled qudits within the OIC (i.e., without external phase modulation) will reduce the overall processing losses. Such designs will allow for more effectively generated keys, particularly for the phase basis (see Supplementary Information S1).

Q3. *The authors show that the phase basis measurement has an additional 17.5 dB of loss, which is a quite severe mismatch with respect to the other basis. Did the author consider how this could affect the security of the protocol? And how does this affect performance? I would expect that the raw key rate, produced in the time-basis, is not affected. However, in the finite-key regime, having a very inefficient phase-basis measurement would force to unbalance the basis choice significantly, reducing consequently the raw key generation rate.*

A3. We thank the Reviewer for this constructive comment and for the suggestion to consider the effects on the raw key generation rate in the finite-key regime. Based on that, we re-analyzed the experimental data and the simulations under the finite-key estimation. In both the experiments and in the simulations, the final secret keys are extracted from the raw keys of the time basis. To evaluate the QBER, we now utilize 4% of the raw keys for the time basis measurements, while we use all keys for the phase basis measurements to compensate for the higher losses due to the measurement setup.

To answer the Reviewer’s comment, we have added a new section in the Supplementary Information, named “Security bound for the BBM92 protocol with time-bin entangled ququarts”, in which we provide the theoretical and experimental details of the security bound for our QKD demonstration considering the finite-key estimation. The added section is reported below.

S2.4 Security bound for the BBM92 protocol with time-bin entangled ququarts

Following standard security definitions⁴, the QKD protocol is stated as ε -secure if it is both ε_{sec} -secret and ε_{cor} -correct. The protocol is called ε_{sec} -secret if the joint state of the output secret key (e.g., on Alice’s side) and of total information from the adversary (e.g., Eve) is statistically indistinguishable from the ideal output state except from some small probability ε_{sec} . The ideal output state is an output key that is uniformly random (in the key space) and completely independent of the adversary’s total information. The protocol is called ε_{cor} -correct if the output secret keys on Alice’s and Bob’s sides are identical except from some small probability ε_{cor} .

The starting point of the security analysis is to ask how many secret bits X can be extracted from Alice’s raw key given Eve’s total information on the QKD system, denoted as E . To this end, we use the quantum leftover-hash lemma⁵ to bound the secret key length l as

$$l = \max_{\beta \in \left(0, \frac{\varepsilon_{\text{sec}}}{2}\right]} \left[H_{\min}^{\frac{\varepsilon_{\text{sec}}}{2} - \beta} (X|E) + 4 \log_2 \beta - 2 \right], \quad (\text{S.21})$$

where $H_{\min}^{\frac{\varepsilon_{\text{sec}}}{2}-\beta}$ is the smooth min-entropy of X given E , and β is a constrained optimization parameter (more details can be found in Ref.⁵). Then, we use the entropic uncertainty relations for qudits to bound the smooth min-entropy

$$H_{\min}^{\frac{\varepsilon_{\text{sec}}}{2}-\beta}(X|E) \geq n \left[\log_2 d - H_d(e_f + \delta(n, k, \beta)) \right] - leak_{\text{EC}} - \log_2 \frac{2}{\varepsilon_{\text{cor}}}, \quad (\text{S.22})$$

with

$$leak_{\text{EC}} = 1.2nH_d(e_t). \quad (\text{S.23})$$

In this expression, e_t (e_f) is the QBER of the time (phase) basis, $leak_{\text{EC}}$ is the information leakages for error correction, $\log_2 2/\varepsilon_{\text{cor}}$ is the number of bits published during error verification, and $H_d(x)$ is the d -level Shannon entropy function: $H_d(x) = -x \log_2(x/(d-1)) - (1-x) \log_2(1-x)$.

The term $\delta(n, k, \beta) = \sqrt{(n+k)(k+1) \ln(2/\beta) / nk^2}$ is the noise due to finite-key statistics of the number of raw keys used for parameter estimation k and the raw keys left for key generation n . It quantifies how the data subset used for parameter estimation well represents the entire dataset. We note that $\delta(n, k, \beta)$ goes closer to zero when both n and k become larger. We thus calculate the lengths l of the keys through the expression

$$l = \max_{\beta \in (0, \varepsilon_{\text{sec}}/4)} \left[n \left[\log_2 d - H_d(e_f + \delta(n, k, \beta)) \right] - leak_{\text{EC}} - \log_2 \frac{8}{\beta^4 \varepsilon_{\text{cor}}} \right]. \quad (\text{S.24})$$

In the experiment and simulation presented in the main text, we assume that $\varepsilon_{\text{sec}} = 10^{-9}$ and $\varepsilon_{\text{cor}} = 10^{-12}$. As shown in Fig. 4b and 4c of the main text, each data point represents 10 minutes of acquisition time, leading to an average finite-key statistical noise of 1.32% (we made the QKD experiment run for 5 hours). Fig. 4d of the main text reports the SKR versus the channel loss. Experimental data were acquired with a collection interval ranging from 30 to 90 minutes, while the simulated results are obtained by assuming raw key lengths after 1-hour acquisition.

In addition, we have revised the main text by reporting the secret key rates obtained in the finite-key regime analysis as follows:

Our QKD experiment ran for five hours, yielding an average QBER of 10.97% and 13.05% for the time and the phase basis, respectively. The obtained QBERs are well within the threshold of

18.93% tolerated by standard BB84 protocols based on ququarts⁵ (see Fig. 4b and Methods). Over the same time lapse, we measured an average secret key rate of 2.04 kbit/s, which was determined through finite-key analysis (see Fig. 4c, Methods and Supplementary Information S2.4). The QKD scheme presented in our work can potentially enable secret key rates of 102.8 kbit/s by simply replacing the SNSPDs with state-of-the-art detectors⁵⁹ featuring lower jitter times (e.g., < 30 ps), see Fig. 4d and Supplementary Information S2.2 and S4.

Finally, we extended the QKD experiment over a 60-km long telecommunication fiber link (see Supplementary Information S2.5). From this, we measured quantum interference after entangled state transmission yielding a raw visibility of 89.34%±2.55% and a secret key rate of 37 bits/s (potentially increasable to 1.93 kbit/s when suppressing the losses of the external temporal gating system). To demonstrate the enhancement in the effective key rates achievable via 4-level entangled qudits with respect to their 2-level counterparts, we reproduced the BBM92 protocol with entangled qubits (see Supplementary Information S2.5). Results showcasing such an enhancement are illustrated in Fig. 4d.

We have also included this analysis at the end of the Methods section “BBM92-like protocol with time-bin entangled photonic ququarts”.

...From 4-level quantum interference, we measured a visibility of 90.14%±1.08% and thus a $\lambda_d = 17.95\%$ based on Eq. (5), from which we estimated a phase basis error of 13.46%, matching our QKD results. The secret key length l was calculated through the expression^{3,71}

$$l = \max_{\beta \in (0, \varepsilon_{\text{sec}}/4)} \left[n[\log_2 d - H_d(e_f + \delta)] - \text{leak}_{\text{EC}} - \log_2 \frac{8}{\beta^4 \varepsilon_{\text{cor}}} \right]. \quad (9)$$

where n is the sifted key rate, e_t (e_f) is the measured QBER of the time (phase) basis, δ is the statistical noise due to finite-key statistics, leak_{EC} is the information leakages for error correction and $H_d(x)$ is the d -level Shannon entropy function: $H_d(x) = -x \log_2(x/(d-1)) - (1-x) \log_2(1-x)$. β is a constrained optimization parameter and we set $\varepsilon_{\text{sec}} = 10^{-9}$ and $\varepsilon_{\text{cor}} = 10^{-12}$ (see Supplementary Information S2.4).

We have revised Fig. 4 of the main text.

Revised figure and caption in the main text. Figure 4. BBM92-like scheme using time-bin entangled ququarts ($d=4$) and comparison with qubits ($d=2$). **a)** Simplified scheme showing the transmission of the signal photon to Alice's setup and the idler photon to Bob's setup for time and phase basis measurements. For simplicity, only one of the measurement setups is shown. The photons passed through a beam splitter to randomly experience either the time or the phase basis. For time basis measurements, we projected the qudits into one of the time bins by direct detection. For phase basis measurements, we projected the qudits into one of the phase vectors by using a phase modulator and the OIC. **b)** QBER of time (phase) basis with an overall average error of $e_t = 10.97\%$ ($e_t = 13.05\%$). **c)** Secret key rates measured over 5 hours, showing an overall average key rate of 2.04 kbit/s. Each data point represents 10 minutes of acquisition time. **d)** Secret key rates for (blue) 4-level and (orange) 2-level QKD versus channel loss (dB). The diamond and square markers show the experimental data acquired for entangled-ququart and entangled-qubit QKD schemes, respectively, when external temporal gating was applied. The circle markers represent the experimental data for entangled-qubit QKD without external temporal gating (w/o gate), i.e., without the loss from this system. This enhances the key rates by a factor of ~ 40 , which matches the estimated efficiency of 2.51% for the external temporal gating system (see Supplementary Information S2 and S4). The blue and red star markers represent experimental data when 60 km of fiber (17 dB loss) is added to the system. The dashed and solid lines show simulation results. PM: phase modulator, OIC: on-chip interferometer cascade.

We finally revised section S1.1 of the Supplementary Information to better explain the setup utilized for quantum state processing and phase basis measurement in the QKD experiment.

S1.1 Experimental scheme for quantum state processing

The setup used for quantum state processing and phase-basis measurement in the QKD experiment had a total transmission loss of ~ 17.5 dB, which included ~ 5 dB for the programmable filter, ~ 3 dB for the phase modulator, ~ 7.5 dB for the OIC (4.5 dB insertion loss and 3 dB splitting loss), and ~ 2 dB for the fiber components (polarization controllers and polarization beam splitters, used for polarization alignment). Phases to qudit states were applied through external phase modulation, which was implemented by using the programmable filter, the electro-optic phase modulator, and the arbitrary waveform generator. First, we optimized the system by making the classical pump burst interfere, and second, we processed the generated photonic qudits. Both classical and quantum interference measurements could be achieved without changing the setup, by simply

adjusting the spectral bands of the programmable filter. For system optimization, the programmable filter was set in such a way to let pass the pump pulses toward the phase modulator and the OIC. This way, we could straightforwardly align the timing between electronic and optical signals through monitoring the interference of the temporally delayed classical pump pulses using an ultrafast photodiode (50 GHz bandwidth) and an electronic sampling scope (Tektronix, CSA8200, 76 GHz bandwidth). Upon successful alignment, we then switched the programmable filter to block the pump's band and let pass the signal and idler photons' bands only, for quantum state processing. We note that future OIC designs could involve active, thermal phase controls on the unbalanced Mach-Zehnder interferometers (MZIs) to ensure full quantum state processing within the OIC without external phase modulation. This would allow removing the electro-optic phase modulators, the programmable filter, and some of the fiber components, thus reducing the total losses of the setup by ~ 9 dB, with the overall loss of the processing system at ~ 8.5 dB.

We thank the Reviewer again for all the constructive comments on our work, which helped us to carefully revise the manuscript and discuss relevant aspects to improve its quality and content. We have addressed all the comments raised by the Reviewer and we believe that our replies answer all the raised concerns. As such, we hope that we have strengthened the suitability of our manuscript for publication in *Nature Communications*, while convincing the Reviewer of both the quality and importance of our work.

Reviewer #2 (Remarks to the Author):

The authors present a time-bin entanglement qudits generation based on an integrated photonic platform, achieving a high quantum information density. They also demonstrate an implementation of the BBM92 QKD protocol using the 4-level qudit entanglement. This work is impressive in its high technical quality. The multi-level entanglement generation method and the results are instructive, and its potential applications can attract broad interest in the field.

We deeply thank the Reviewer for pricing the high technical quality of our work and for acknowledging its potential applications and broad interest in the field.

I will consider its publication in a high impact journal, like Nature Communications, once the following comments are adequately addressed.

We sincerely thank the Reviewer for considering the publication of our manuscript upon addressing specific comments that we have fully considered and, we believe, addressed in the answers provided below.

Q1. *The power and repetition rate of the pump laser, as well as the brightness of the entanglement source, should be provided in the main text for a better understanding of the source.*

A1. We thank the Reviewer for this comment. Throughout the experiment for quantum state characterization, the repetition rate of the laser was fixed at 250 MHz, with a peak power for each pump pulse of ~6.5 W. For QKD demonstrations, the repetition rate was increased to 1 GHz, with the pump peak power being the same at ~6.5 W. The brightness of the entanglement source (the spiral waveguide) is 1.018×10^{-12} pairs/(mW²·GHz) per pulse.

As per Reviewer's suggestion, we have added these values in the main text.

... We exploit the OIC as a multi-path optical splitter to prepare a pump pulse sequence (at a repetition rate of 250 MHz, corresponding to a 4-ns delay between pump burst sequences, see Methods) for the generation of time-bin entangled qudits.

To prove the suitability of our photonic platform for QKD applications, as well as an increase in the effective bit rate capacity, we implemented a BBM92-like scheme extended to ququarts. Benefitting from the ultrafast nature of the time-bin qudits enabled by the OIC, we could implement the QKD protocol by further increasing the repetition rate of the laser to 1 GHz,

corresponding to a 1-ns delay between pump burst sequences (see Methods). Specifically, we implemented a BBM92-like protocol in which two clients...

...In our work, this limitation comes from the noise associated to the spiral waveguide used as entanglement source, which can be improved significantly by dedicated source engineering. Our spiral operates at a brightness of 1.018×10^{-12} pairs/(mW²·GHz) per pulse and at a coincidence-to-accidental ratio (CAR) of 20.

Furthermore, based on the Reviewer's comment, we have moved the description of the experimental setup from section S1.1 of the Supplementary Information to the Methods in such a way to provide in the Article file a clearer description of the experimental setup.

The Methods section reads as follows:

Experimental setup. The experimental setup utilized for entangled photon generation and processing is reported in Fig. S1 of the Supplementary Information. The OIC and a 45cm-long spiral waveguide were fabricated on a single CMOS-compatible photonic chip made of high-index doped silica³⁶. The fiber-pigtailed OIC featured an overall input-to-output loss of < 4.5 dB. The *d*-fold pump pulse train was prepared from a mode-locked laser (Menlo Systems Inc., FC1500-250-WG, 250 MHz repetition rate), which was spectrally filtered to ~5 ps pulse duration centered at 1,556.15 nm (corresponding to the H26 telecom channel). The generated pulses were amplified with an erbium-doped fiber amplifier (EDFA, Keopsys, CEFA-C-BO-HP) and then launched into the spiral waveguide to induce spontaneous four-wave-mixing (SFWM) for the generation of signal and idler photon pairs. Photons were then directed into a programmable filter (Finisar WaveShaper 4000A), an electro-optic phase modulator (EO-Space, PM-5VEK-40-PFA-PFA-UV-UL), and then back into the OIC for quantum state processing.

After coherent mixing of the time bins in the OIC, the signal and idler photons were separated into two fiber channels using a standard wavelength-division multiplexer (LightWave2020). After demultiplexing, we used two 40 GHz intensity modulators driven by a 62.5 GSa/s AWG [arbitrary waveform generator] (Keysight) to implement 25ps-wide of temporal gating² for the signal and idler photons. The photons were then directed into two SNSPDs [superconducting nanowire single-photon detectors] (Quantum Opus, 85% photon efficiency at 1,550 nm, ~52 ps jitter time,

and 20 ns dead time). We measured photon coincidences by using a time tagging electronic unit (PicoQuant HydraHarp).

We note that the repetition rate of the mode-locked laser was increased to 1 GHz for the QKD experiment. To this end, we made use of a system comprising three polarization maintaining (PM) beam splitters (BSs) and short PM fibers of different lengths. Between the first and the second BSs, the length difference of the two fibers is 2 ns, thus doubling the repetition rate to 500 MHz. Then, between the second and the third BS, the length difference between the fibers is 1 ns, thus doubling the repetition rate from 500 MHz to 1 GHz.

The revised Section S1.1 of the Supplementary Information can be found at the end of A3 that we provide to Q3 raised from Reviewer #1.

***Q2.a.** Is the finite-key effect taken into consideration when calculating the experimental key rates? Note that this is an important part for QKD.*

A2.a. We thank the Reviewer for this insightful comment. Based on that, we re-analyzed both the experimental data and the simulations under the finite-key estimation. We would like to acknowledge that Reviewer #1 asked a very similar question. Hence, we kindly ask the Reviewer to also refer to A3 that we have provided to Reviewer#1, as well as to the revised part in the main text, Methods, and Supplementary Information reported herein.

***Q2.b.** Moreover, the reported QBER of 11.98% is much higher than standard BB84. What are the limiting factors on this high QBER? On this point, a recent review on the subject of QKD will be helpful for readers, e.g., [Reviews of Modern Physics 92, 025002 (2020)].*

A2.b. We thank the Reviewer for this comment and for suggesting the review paper, cited as Ref. [58] in the revised manuscript. As the Reviewer pointed out, standard BB84 protocols based on qubits can tolerate QBERs up to 11%, while QKD schemes based on 4-level qudits can increase such tolerance up to 18.93% (*Phys. Rev. Lett.* **88**, 127902 (2002) and *Phys. Rev. Appl.* **18**, 044027 (2022), cited as Refs. [16] and [67] in the revised manuscript, respectively). For this reason, in the original manuscript, we found it more reasonable to directly compare the measured QBER with the QBER associated to 4 levels (18.93%). Yet, we understand the Reviewer's concern. The main source of noise in the BBM92-like scheme proposed here comes from the spiral waveguide utilized for the generation of entangled photon pairs. The coincidence-to-accidental ratio (CAR) of the spiral waveguide is affected by undesirable spontaneous Raman scattering photons and multi-photon events.

To answer the Reviewer's comment, we have added a new section in the Supplementary Information, named "Characterization of the spiral waveguide", in which we provide a detailed theoretical and experimental study of the spiral waveguide, along with simulations to show its impact in our QKD implementation, including the 13.05% phase error (from further calculation made during the revision process). We have also added a discussion in the revised main text with respect to this aspect. We kindly ask the Reviewer to refer as well to A2, that we have provided to Reviewer#1, raising a similar question.

Q3. *A recent work on frequency-bin qudit entanglement generation [Phys. Rev. Appl. 19, 064026 (2023)] has a frequency bin spacing of 15 GHz. The authors may add this reference to the manuscript and include it in the entanglement source comparison in Figure 1c.*

A3. We thank the Reviewer for bringing this interesting work to our attention. There, the authors achieve a very high time-bandwidth product, benefiting from the narrow spacing between the frequency bins (15 GHz) and the time duration (~ 0.89 ns) of the generated entangled photon pairs. The time duration provided in that work is given by the -30dB full temporal width of the Lorentzian lineshape calculated based on the Q factor of the microring resonator. As per Reviewer's suggestion, we have properly referenced this work (cited as Ref. [43] of the revised manuscript) and used it in the entanglement source comparison in Figure 1c (the work is marked as "Borghini, 2023").

The revised main text reads:

We note that a similar quantum information density of 0.148 has recently been achieved for 4-level states, by leveraging a frequency comb spacing as narrow as 15 GHz (yet, within hundreds of ps-large temporal width⁴³), see Fig. 1b,c for further details^{2,19,22,23,41-45}.

Revised Fig. 1c is reported below.

Revised figure. Figure 1 c) Current state-of-art in scaling the photonic Hilbert space size per TBP of two-photon sources using time-bin and/or frequency-bin entanglement. A decrease can be noted as the photon state dimensionality increases for sources highlighted in gray, which is intrinsic to current approaches **dealing with** two-photon frequency combs. The full temporal width of the frequency modes generated from microring resonators are evaluated by their Q-factors and spectral line widths. The compactness of the time-bin entangled states enabled through our platform allows to bypass this limitation (red symbols).

Q4. Elaborate more clearly on the difference between the solid and dashed lines in Figure 4d, as well as the experimental setup corresponding to the blue and red stars. The data points for the ‘ideal’ case may be removed, because this is not the real data in your experiment. The solid lines are sufficient to show the expectation.

A4. We thank the Reviewer for this comment. We agree that, in the original manuscript, we did not comprehensively elaborate on the content of Fig 4d. We also understand that the definition of “ideal” was somehow misleading in the original manuscript. By “ideal”, we meant without applying the external temporal gating, as opposed to the experiment, in which we did apply the gating. For the sake of clarity, we have revised the legend of Fig. 4d and replaced “ideal” with “w/o gating”. We understand the Reviewer’s suggestion to remove the data points related to the “ideal” cases. Yet, as detailed in the following, we believe it is better to keep them for a more comprehensive analysis of our QKD scheme. We hope that, after our explanation and revision of the figure, the Reviewer will be convinced that these points should be kept.

Here, we report the revised figure and the caption, together with the experimental setup utilized for the related measurements. In the dashed blue lines (color associated with 4-level qudits), the diamond markers represent the experimental data acquired when the external temporal gating was applied (as discussed in the manuscript, this gating was necessary to bypass the issue related to the jitter time of the SNSPDs (~52 ps) approaching the time mode spacing (64 ps), and thus to improve the resolution of the time bins). The temporal gating was implemented by making use of two

intensity modulators (IMs in revised Fig. S3), one in the time basis and one in the phase basis measurement setup. A variable optical attenuator (VOA in revised Fig. S3) was used to introduce losses into the signal and idler photons' channel prior to the beam splitter-based delay scheme. In order to make a fair comparison between the BBM92 scheme obtained with 4-level (ququart) and 2-level (qubit) entangled photons, we utilized the same experimental setting (i.e., IMs for temporal gating + VOA to introduce losses) also for qubits. Data acquired in this setting are represented by the square markers in the dashed orange line (color associated with qubits).

The solid lines instead represent the simulations in which no temporal gating is applied, i.e., in the absence of the losses from the two IMs. The blue color is associated with ququarts, while the orange color is associated with qubits. Since the time mode spacing of the qubits (128 ps) was much larger than the jitter time of the SNSPDs (52 ps), we could acquire experimental data without the two IMs, which are shown by those circle markers in the solid orange line. The blue star represents measurements acquired for entangled ququarts by replacing the VOA with a 30 km-long fiber system where each photon propagates, while still applying the external temporal gating (necessary to resolve 64-ps spaced qudits). The red star represents measurements acquired for entangled qubits by replacing the VOA with a 30 km-long fiber system and by removing the two IMs.

Based on the Reviewer's comment, we have revised Fig. 4d and the related caption, as follows:

Revised figure and caption. Figure 4 d) Secret key rates for (blue) 4-level and (orange) 2-level QKD versus channel loss (dB). The diamond and square markers show the experimental data acquired for **entangled-ququart and entangled-qubit QKD schemes**, respectively, when external temporal gating was applied. The circle markers represent the experimental data for **entangled-qubit QKD without external temporal gating (w/o gate)**, i.e., without the loss from this system. This enhances the key rates by a factor of ~ 40 , which matches the estimated efficiency of 2.51% for the external temporal gating system (see Supplementary **Information S2 and S4**). The blue and red star markers represent experimental data when 60 km of fiber (17 dB loss) is added to the system. The dashed and solid lines show simulation results. PM: phase modulator, OIC: on-chip interferometer cascade.

We have also modified some parts in sections S2.2 and S2.3 of the Supplementary Information to better describe the experimental setup associated to the QKD experiment.

S2.2 Implementation of the BBM92 protocol with time-bin entangled ququarts ($d=4$)

The 64 ps spacing of the time bins almost approached the ~ 52 ps jitter time of the SNSPDs used in our experiment. While such a jitter time can still resolve 4-level qudits, it brings a higher crosstalk from adjacent time bins, yielding an increase of the **quantum bit error rate (QBER, see section S4)**. To address this issue, we implemented external temporal **gating, which was experimentally implemented by making use of two intensity modulators (IMs in Fig. S3), one in the time basis and one in the phase basis measurement setup. A variable optical attenuator (VOA in Fig. S3) was used to introduce losses into the signal and idler photons' channel prior to the beam splitter-based delay scheme.**

In the time basis measurement system, the temporal gating was applied to select only modes...

In the phase basis measurement system, we applied external phase modulation to project the signal and idler photons into different phase vectors. The external temporal gating was then used to select the superposition state...

S2.3 Implementation of BBM92 scheme with time-bin entangled qubits ($d=2$)

For this **purpose**, we generated 2-level photon pairs featuring a time bin spacing of 128 ps, the resolution of which was not affected by the SNSPD jitter time. Still, to make a fair comparison between the QKD schemes based on 2- and 4-level entangled states, **we utilized the same experimental setting (i.e., IMs for temporal gating + VOA to introduce losses) also for qubits. This gating only selected the time mode...**

The comparison between ququart- and qubit-based BBM92 schemes is illustrated in Fig. 4d of the main text. The solid lines represent the cases in which no temporal gating is applied, i.e., in the absence of the two IMs. Blue and orange colors are associated with ququarts and qubits, respectively. The blue star represents measurements acquired for entangled ququarts by replacing the VOA with a 30 km-long fiber system, while still applying the external temporal gating (necessary to resolve 64-ps spaced qudits). Instead, since the time mode spacing of the qubits (128 ps) was much larger than the jitter time of the SNSPDs (~ 52 ps), we could attain experimental data without recurring to the two IMs. The red star represents measurements acquired for entangled

qubits by replacing the VOA with a 30 km-long fiber system where each photon propagates and by removing the two IMs (see section S2.5). The implementation of qubit-based BBM92 QKD without external temporal gating led to an enhancement of the secret key rates (SKRs) by a factor of ~ 40 compared to the case in which the external temporal gating was applied (see Fig. 4 of the main text). This result agreed with the estimated efficiency of 2.51% for the temporal gating.

We have added the VOA in Figure S3 of the Supplementary Information and modified the related caption:

Revised Figure and caption. Figure S3. Setup for BBM92-like QKD based on time-bin entangled states. Following the photon pair generation stage, a variable optical attenuator was used to introduce losses into the signal and idler photons' channel prior to the fiber-based delay system in order to simulate transmission losses. A fiber-based delay system was used to reproduce the random choice of the mutually unbiased bases by Alice and Bob. Photons are randomly directed into either the time- or phase-measurement setups through two 50:50 beam splitters, either at the same time or with a 500 ps temporal delay. Time basis measurements were achieved by applying temporal gating to photons, while phase basis measurements were carried out by applying electro-optic phase modulation and then temporal gating on the time bin superposition. Such temporal gating was implemented by making use of two intensity modulators, one in the time basis and one in the phase basis measurement setup. The sifted keys were collected from the coincidence events obtained from the time basis measurements, while the security of the system was monitored through the bit error rates from both the time and phase basis measurements. TDL: tunable delay line, OIC: on-chip interferometer cascade, EDFA: erbium-doped fiber amplifier, VOA: variable optical attenuator, PM: phase modulator, IM: intensity modulator, DEMUX: demultiplexer, PC: polarization controller, BPF: band pass filter, BS: beam splitter, PF: programmable filter, NF: notch filter, SNSPD: superconducting nanowire single-photon detector.

Q5. The authors report a key rate of 2.47 Kbps or potentially improve to 100.9 Kbps. But this is much lower than state-of-art real-time key rate of >110 Mbps [Nature Photonics 17, 416 (2023)]. One motivation of high-dimensional QKD is the potential of high secret rates. Unfortunately, this experiment does not prove this potential. Also, the reported rate here is not real time yet. The authors should carefully discuss this point and the application front of high-dimensional QKD on high key rates.

A5. We thank the Reviewer for this comment, which we carefully considered in revising the manuscript. Such high real-time key rates as those mentioned by the Reviewer have been reached so far through approaches such as prepare-and-measure protocols. The scheme presented in our manuscript is intrinsically different from these works, as it is based on an entanglement source. We agree that entanglement-based protocols, even when scaled to qudits, cannot achieve real-time key rates as high as prepare-and-measure counterparts. In time-bin encoding schemes, the reachable effective key rate is mainly limited by the repetition rate of the system and by the long dead time of single-photon detectors (tens of nanoseconds).

In entanglement-based QKD, another important limiting factor is associated to the rate at which correlated photon pairs are probabilistically generated from the entanglement source. This is especially the case for entangled qubits. The effective key rate is affected by the fact that a certain percentage of detected photon events from the entanglement source become uncorrelated due to losses and noise present in the system. In this scenario, two strategies could be used to increase the key rate. One approach could be to utilize an entangled-photon source operating at a higher mean photon pair generation rate. However, due to the probabilistic nature of parametric two-photon sources, a higher mean photon number requires higher pump power, resulting in more multiphoton emissions and increased linear noise generation. This, in turn, results in a worse noise profile and lower coincidence-to-accidental ratios, which make the system unsuitable for reliable QKD experiments. The other approach would be to increase the bit capacity per photon pair coincidence. For a given entangled-photon source (operating at the same mean photon pair generation rate) and channel loss, the coincidence events per second of the source is almost fixed. If each event is encoded with a higher number of bits, such as in qudit-based protocols and in our scheme, the effective QKD key rate could be increased.

With our work, we want to show that the on-chip interferometric system proposed for implementing the BBM92 protocol can, to some degree, compensate for these drawbacks by enabling secret key rates that are considerably higher than those registered so far with entangled qubits. This is why we compared the QKD scheme achievable with ququarts and qubits at the same experimental conditions. Indeed, despite the relatively low performance of our entanglement source, we consider the effective key rate of 2.04 kbit/s and the extendable distance of 60-km over optical fiber link are quite noteworthy when comparing to entangled qubit demonstrations.

Our experiment proves the potential of on-chip interferometric systems towards benefitting of all the advantages brought by qudits also in entanglement-based protocols. This assumes a further value when considering the ultrafast nature of the entangled qudits, which can be scaled up without sacrificing the repetition rate of the system, as well as their processing at telecommunication speeds. Future integrated waveguide technologies enabling, for instance, phase modulation of the photonic qudits within the same chip, combined with better entanglement sources, can open new fronts in the application of entangled qudits in QKD implementations.

Based on the Reviewer's comment, we added a discussion in the conclusion of the manuscript, which reads:

The maximum accessible number of time bins in this platform is 256, which can potentially enable, alongside **with** using a sub-ps pulsed pumping scheme, the generation of 256-level time-bin entangled photonic qudits with a time bin spacing of 1 ps. **The OIC hence allows for scaling up the qudit dimensionality to 2^N time bins ($N \leq 8$) under the fixed two-photon temporal width of 256 ps, thus keeping a potential maximum repetition rate of 3.9 GHz.**

The secret key rates reported in this work still remain orders of magnitude below those demonstrated via, e.g., prepare-and-measure protocols, which can register real-time keys >100 Mbit/s⁶⁰. An important factor that intrinsically limits the performance achievable with entangled photons is associated to the rate at which correlated photon pairs are probabilistically generated from the entanglement source. In our work, this limitation comes from the noise associated to the spiral waveguide used as entanglement source, which can be improved significantly by dedicated source engineering. Our spiral operates at a brightness of 1.018×10^{-12} pairs/(mW²·GHz) per pulse and at a coincidence-to-accidental ratio (CAR) of 20. Noise from the source comprises undesirable spontaneous Raman scattering photons and multi-photon events (see Supplementary Information S3). Yet, despite the relatively low performance of our spiral waveguide (which could be improved, in the longer term, with *ad hoc* fabrication techniques), the effective key rate extracted from the QKD experiment is significant when compared to entangled qubit demonstrations^{11,61,62}. The main scope of this work is indeed to show that the proposed on-chip interferometric framework has the potential to compensate some of the limits associated with entanglement sources and enable high secret key rates also in entanglement-based QKD protocols. On-chip waveguide sources made from other materials offer higher brightness, less or restricted Raman gain, and thus CAR values that are orders of magnitudes higher than ours^{63–65}. Such sources can improve our system by enabling photon pair generation rates in the order of a few MHz under significantly lower pump peak powers (a few mW). These conditions can also reduce uncorrelated and multi-photon events, ultimately improving the security of the QKD scheme. Furthermore, future OIC designs on electro-active material platforms, involving active phase controls on the unbalanced MZIs to fully process the entangled qudits within the OIC (i.e., without external phase modulation) will reduce the overall processing losses. Such designs will allow for more effectively generated keys, particularly for the phase basis (see Supplementary Information S1.1).

Q6. Throughout the manuscript, the timing jitter of the SNSPDs appears to be a main obstacle for characterizing the entangled qudits with an ultrashort time-bin spacing and further increasing the key rates in the implementation of BBM92 QKD. Please consider discussing how the performance of the SNSPDs will affect the qudit entanglement sources and their applications, which can be helpful for a broader readership.

A6. We thank the Reviewer for this observation about the timing jitter of the SNSPDs and for the related suggestion. While the jitter time of our SNSPDs represents an obstacle to resolving ultrafast qudits, we agree that, with the advent of state-of-the-art SNSPDs featuring picosecond-level jitter times (*Optics Express* **27**, 26579-26587 (2019); *IEEE Journal of Selected Topics in Quantum Electronics* **28**, 1-8 (2022); *ACS Photonics* **7**, 1780–1787 (2020), cited as Refs. [66], [67], and [64], respectively), it will become possible to resolve narrow-spaced time bins. This advancement will further increase the dimensionality and bit rate achievable via time-bin qudits in ultrafast implementations, thereby enabling higher secret key rates. Additionally, if combined with on-chip interferometric systems as the one presented in our work, the resolution of narrow-spaced entangled time bins can lead to an enhancement in the bit capacity within a predefined temporal width of the photon state, without affecting the repetition rate of the system. In this regard, we invite the Reviewer to read A1 that we provide to Reviewer#1's Q1.

More broadly, reducing the jitter time will significantly benefit entanglement-distribution quantum communication protocols (beyond QKD) that encode information into the time-of-arrival of photons (i.e., energy-time and time-bin entanglement). Protocols based on time-entanglement, both in their continuous and discrete forms, are constrained by the large jitter times of single-photon detectors when aiming to maximize the bit rate for ultrafast implementations. Yet, we note that, in energy-time entanglement implementations (e.g., *Phys. Rev. Lett.* **98**, 6, 060503 (2007), *Phys. Rev. X* **13**, 021001 (2023), and *New J. Phys.* **17**, 022002 (2015) – cited as Refs. [24], [25], and [26] of the revised manuscript), time bins are generated during quantum state processing by discretizing the temporal shape of the generated photons into time frames, which are further discretized into time bins defining the time-of-arrival of correlated photons. These systems require the optimization of both time bins and frame sizes to minimize error rates due to jitter times. This is not the case in our system and, more in general, in time-bin entanglement implementations, where the times-of-arrival of the entangled photons are predefined since the entanglement source. In these implementations, error rates can be minimized through temporal post-selection.

To provide more details on the performance of the single-photon detectors in our system, we have performed a detailed characterization of the SNSPDs, focusing on the noise introduced by their jitter time for both single-photon and correlation measurements. Initially, we verified that the jitter time of the SNSPDs used in our experiment has a full width at half maximum (FWHM) of ~ 52 ps. For the detection of time-bin ququarts, as illustrated in Fig. S7a (from the revised Supplementary Information), the single-photon histogram demonstrates that the detector jitter time can still resolve the 64 ps-spaced time bins. However, such a limitation in the resolution introduces significant

crosstalk from adjacent time bins, thereby increasing noise in the correlation measurements of the entangled photon states, as shown in the coincidence matrix in Fig. S7b. The overall QBER for these correlation measurements is 23.40%, significantly higher than the threshold set for 4-level QKD (18.93%). To address this issue, we applied external temporal gating, as depicted in Fig. S7c and d, which enabled a time-bin spacing of 128 ps, fully resolvable by the SNSPDs. The coincidence matrix for the entangled ququarts with the external temporal gating system is shown in Fig. S7e, resulting in an overall QBER reduction to ~11%. Additionally, as illustrated in Fig. S8 (from the revised Supplementary Information and reported herein), we simulated the estimated QBER of our 64 ps-spaced entangled ququarts versus the jitter time of SNSPDs. Using SNSPDs with a lower jitter time (i.e., < 30 ps) can significantly reduce the detection crosstalk of adjacent time bins. This will make it possible to remove the external temporal gating system, and thus enable the SKR achievable for 4-level QKD following the solid blue line shown in Fig. 4d of the main text.

Based on the Reviewer's comment, we have added the following discussion in the conclusion of the manuscript.

While the jitter time of our single-photon detectors was an obstacle to resolving the ultrafast qudits, with the advent of state-of-the-art SNSPDs featuring picosecond-level jitter times^{59,66,67}, it will become possible to resolve narrow-spaced time bins. More in general, a variety of quantum communication protocols based on time-entanglement distribution will benefit from reducing the jitter times of the detectors. Protocols based on time-entanglement, both in its continuous and discrete forms, are constrained by the large jitter times of single-photon detectors when aiming to maximize the bit rate for ultrafast implementations. For instance, energy-time entanglement implementations²⁴⁻²⁶, where the photons' times-of-arrival are defined via discretization into time frames and bins during quantum state processing, necessitates the optimization of both time frames and bins to minimize error rates due to jitter times. Yet, we note that this is not the case for time-bin entanglement implementations²¹⁻²³, where the photons' times-of-arrival are predefined since the entanglement source. In these schemes, error rates can be minimized through temporal post-selection, as done in our work.

The advancement in reducing jitter and dead times in single-photon detectors will lead to an increase in the dimensionality and bit rates achievable via time-bin qudits for ultrafast implementations based on single- and entangled-photons. If combined with on-chip interferometric systems as the one presented in our work, resolving narrow-spaced entangled time bins can ultimately lead to an enhancement of the bit capacity within a predefined temporal width

of the photon states, without scarifying the repetition rate of the system. Our work further enables the telecom-ready usage of quantum information processing and demonstrates the significant potential of time-bin entangled photonic systems to achieve high data rates and security levels for a variety of quantum communication protocols in optical fiber links over long distances.

We have also added a new section in the revised Supplementary Information, in which we provide a detailed analysis of the effects of the jitter time on the demonstrated QKD scheme.

S.4. Characterization of the jitter time of the SNSPDs for the detection of time-bin entangled ququarts

The 64 ps spacing of the time bins almost approached the ~ 52 ps jitter time of the SNSPDs used in our experiment. As shown in Fig. S7a, the single-photon histogram indicates that, even though the detector jitter time can still resolve ququarts, it brings crosstalk from adjacent time bins. The shadow regions are Gaussian functions obtained from the photon counting histogram of a single pulse, which has a full width at half maximum (FWHM) of ~ 52 ps. The crosstalk can also be visualized by the coincidence matrix among all time bins of the entangled ququarts, as shown in Fig. S7b, which yields a QBER of 23.40%. Such value is beyond the QBER threshold for 4-level QKD. To address this issue, we implemented external temporal gating in the time basis measurement system that selected only modes $|0\rangle$ and $|2\rangle$ for the short path of the beam splitter-based delay scheme, and only modes $|1\rangle$ and $|3\rangle$ for the long path, as shown in Fig. S7c and d, respectively. The external temporal gating system enabled a time-bin spacing of 128ps, which can be fully resolved by the SNSPDs. The coincidence matrix among all time bins of the entangled ququarts with the external temporal gating system is shown in Fig. S7e. The overall QBER is thus reduced to $\sim 11\%$. Such residual QBER originates from the source noise. Furthermore, as illustrated in Fig. S8, we simulated the estimated QBER of our 64 ps-spaced entangled ququarts versus the jitter time of SNSPDs. We note that if SNSPDs with lower jitter time² (i.e., < 30 ps) could be used, the detection crosstalk of the adjacent time bins could be eliminated. This would make it possible to remove the external temporal gating system, and thus achieve the SKR for 4-level QKD predicted by the solid blue line shown in Fig. 4d of the main text (named there as “without gating curve”, w/o in Fig. 4 of the main text).

Figure S7. Analysis of the jitter time of the SNSPDs on the time-bin qudits' resolution. **a)** Single-photon histogram of the 4-level qudits. The shadow regions are Gaussian functions obtained from the photon counting histogram of a single pulse, which has a full width at half maximum (FWHM) of ~ 52 ps. **b)** Coincidence matrix among all time bins of the 4-level qudits. The post-selection time window was 44 ps. **c,d)** Single-photon histograms of the 4-level qudits after applying an external temporal gating. **e)** Coincidence matrix among all time bins of the entangled ququarts, obtained by using the external temporal gating. The post-selection time window was 100 ps.

Figure S8. Simulations of the QBER of 64 ps-spaced entangled ququarts versus the jitter time of SNSPDs. The black triangle is the experimental data obtained without external temporal gating. The post-selection time window was fixed at 44 ps.

Q7. Ref. [3] and [54] are the same.

A7. We corrected the reference in the revised manuscript.

We thank once more Reviewer#2 for the positive feedback expressed about our manuscript and their conditional recommendation for publication *Nature Communications*. We hope to have addressed all the comments and questions raised from the Reviewer, which we found particularly constructive towards improving our submitted manuscript.